# QUANTITATIVELY EVALUATING GANS
# WITH DIVERGENCES PROPOSED FOR TRAINING

**Daniel Jiwoong Im**[1,2]**, He Ma**[3,4]**, Graham Taylor**[3,4]**, & Kristin Branson**[1]
[1]Janelia Research Campus, HHMI, [2]AIFounded Inc. [3]University of Guelph, [4]Vector Institute
`{imd, bransonk}@janelia.hhmi.org`
`{hma02, gtaylor}@uoguelph.ca`

## ABSTRACT

Generative adversarial networks (GANs) have been extremely effective in approximating complex distributions of high-dimensional, input data samples, and substantial progress has been made in understanding and improving GAN performance in terms of both theory and application. However, we currently lack quantitative methods for model assessment. Because ot this, while many GAN variants being proposed, we have relatively little understanding of their relative abilities. In this paper, we evaluate the performance of various types of GANs using divergence and distance functions typically used only for training. We observe consistency across the various proposed metrics and, interestingly, the test-time metrics do not favour networks that use the same training-time criterion. We also compare the proposed metrics to human perceptual scores.

## 1 INTRODUCTION

Generative adversarial networks (GANs) aim to approximate a data distribution $P$, using a parameterized model distribution $Q$. They achieve this by jointly optimizing generative and discriminative networks (Goodfellow et al., 2014). GANs are end-to-end differentiable, and samples from the generative network are propagated forward to a discriminative network, and error signals are then propagated backwards from the discriminative network to the generative network. The discriminative network is often viewed as learned, adaptive loss function for the generative network.

GANs have achieved state-of-the-art results for a number of applications (Goodfellow, 2016), producing more realistic, sharper samples than other popular generative models, such as variational autoencoders (Kingma & Welling, 2014). Because of their success, many GAN frameworks have been proposed. However, it has been difficult to compare these algorithms and understand their strengths and weaknesses because we are currently lacking in quantitative methods for assessing the learned generators.

In this work, we propose new metrics for measuring how realistic samples generated from GANs are. These criteria are based on a formulation of divergence between the distributions $P$ and $Q$ (Nowozin et al., 2016; Sriperumbudur et al., 2009):

$$\inf_Q J(Q) = \inf_Q \sup_{f \in \mathcal{F}} \mathbb{E}_{P(x)}\left[\mu\left(f(x)\right)\right] - \mathbb{E}_{Q(x)}\left[\upsilon\left(f(x)\right)\right] \tag{1}$$

Here, different choices of $\mu$, $\upsilon$, and $\mathcal{F}$ can correspond to different $f$-divergences (Nowozin et al., 2016) or different integral probability metrics (IPMs) (Sriperumbudur et al., 2009). Importantly, $J(Q)$ can be estimated using samples from $P$ and $Q$, and does not require us to be able to estimate $P(x)$ or $Q(x)$ for samples $x$. Instead, evaluating $J(Q)$ involves finding the function $f \in \mathcal{F}$ that is maximally different with respect to $P$ and $Q$.

This measure of divergence between the distributions $P$ and $Q$ is related to the GAN criterion if we restrict the function class $\mathcal{F}$ to be neural network functions parameterized by the vector $\phi$ and the class of approximating distributions to correspond to neural network generators $G_\theta$ parameterized by the vector $\theta$, allowing formulation as a min-max problem:

$$\min_\theta J(\theta) = \min_\theta \max_\phi \mathbb{E}_{P(x)}\left[\mu\left(D_\phi(x)\right)\right] - \mathbb{E}_{Q_\theta(x)}\left[\upsilon\left(D_\phi(x)\right)\right], \tag{2}$$

Table 1: Defined $\mu$ and $\upsilon$ functions for GAN metrics proposed in this paper. $M$ is some real number. $\mathcal{H}$ is a Reproducing Kernel Hilbert Space (RKHS) and $\| \cdot \|_L$ is the Lipschitz constant. For the LS-DCGAN, we used $b = 1$ and $a = 0$ (Mao et al., 2017).

| Metric | $\mu$ | $\upsilon$ | Function Class |
|---|---|---|---|
| GAN (GC) | $\log f$ | $-\log(1 - f)$ | $\mathcal{X} \to \mathbb{R}_+, \exists M \in \mathbb{R} : |f(x)| \leq M$ |
| Least-Squares GAN (LS) | $-(f - b)^2$ | $(f - a)^2$ | $\mathcal{X} \to \mathbb{R}, \exists M \in \mathbb{R} : |f(x)| \leq M$ |
| MMD | $f$ | $f$ | $f : \|f\|_{\mathcal{H}} \leq 1$ |
| Wasserstein (IW) | $f$ | $f$ | $f : \|f\|_L \leq 1$ |

In this formulation, $Q_\theta$ corresponds to the generator network's distribution and $D_\phi$ corresponds to the discriminator network (see (Nowozin et al., 2016) for details).

We propose using $J(\theta)$ to evaluate the performance of the generator network $G_\theta$ for various choices of $\mu$ and $\upsilon$, corresponding to different $f$-divergences or IPMs between distributions $P$ and $Q_\theta$, that have been successfully used for GAN training. Our proposed metrics differ from most existing metrics in that they are adaptive, and involve finding the maximum over discriminative networks. We compare four metrics, those corresponding to the original GAN (GC) (Goodfellow, 2016), the Least-Squares GAN (LS) (Mao et al., 2017), the Wasserstein GAN (IW) (Arjovsky et al., 2017), and the Maximum Mean Discrepency GAN (MMD) (Li et al., 2017) criteria. Choices for $\mu$, $\upsilon$, and $\mathcal{F}$ for these metrics are shown in Table 1. Our method can easily be extended to other $f$-divergences or IPMs.

To compare these and previous metrics for evaluating GANs, we performed many experiments, training and comparing multiple types of GANs with multiple architectures on multiple data sets. We qualitatively and quantitatively compared these metrics to human perception, and found that our proposed metrics better reflected human perception. We also show that rankings produced using our proposed metrics are consistent across metrics, thus are robust to the exact choices of the functions $\mu$ and $\upsilon$ in Equation 2.

We used the proposed metrics to quantitatively analyze three different families of GANs: Deep Convolutional Generative Adversarial Networks (DCGAN) (Radford et al., 2015), Least-Squares GANs (LS-DCGAN), and Wasserstein GANs (W-DCGAN), each of which corresponded to a different proposed metric. Interestingly, we found that the different proposed metrics still agreed on the best GAN framework for each dataset. Thus, even though, e.g. for MNIST the W-DCGAN was trained with the IW criterion, LS-DCGAN still outperformed it for the IW criterion.

Our analysis also included carrying out a sensitivity analysis with respect to various factors, such as the architecture size, noise dimension, update ratio between discriminator and generator, and number of data points. Our empirical results show that: i) the larger the GAN architecture, the better the results; ii) having a generator network larger than the discriminator network does not yield good results; iii) the best ratio between discriminator and generator updates depend on the data set; and iv) the W-DCGAN and LS-DCGAN performance increases much faster than DCGAN as the number of training examples grows. These metrics thus allow us to tune the hyper-parameters and architectures of GANs based on our proposed method.

## 2 RELATED WORK

GANs can be evaluated using manual annotations, but this is time consuming and difficult to reproduce. Several automatically computable metrics have been proposed for evaluating the performance of probabilistic general models and GANs in particular. We review some of these here, and compare our proposed metrics to these in our experiments.

Many previous probabilistic generative models were evaluated based on the pointwise likelihood of the test data, the criterion also used during training. While GANs can be used to generate samples from the approximate distribution, its likelihood on test samples cannot be evaluated without simplifying assumptions. As discussed in (Theis et al., 2015), likelihood often does not provide good rankings of how realistic samples look, the main goal of GANs. We evaluted the efficacy of the log-likelihood of the test data, as estimated using **Annealed Importance Sampling** (AIS) (Wu et al., 2016). AIS

has been to estimate the likelihood of a test sample $x$ by considering many intermediate distributions that are defined by taking a weighted geometric mean between the prior (input) distribution, $p(z)$, and an approximation of the joint distribution $p_\sigma(x, z) = p_\sigma(x|z)p(z)$. Here, $p_\sigma(x|z)$ is a Gaussian kernel with fixed standard deviation $\sigma$ around mean $G_\theta(z)$. The final estimate depends critically on the accuracy of this approximation. In Section 4, we demonstrate that the AIS estimate of $p(x)$ is highly dependent on the choice of this hyperparameter.

The **Generative Adversarial Metric** (Im et al., 2016a) measures the relative performance of two GANs by measuring the likelihood ratio of the two models. Consider two GANs with their respective trained partners, $M_1 = (D_1, G_1)$ and $M_2 = (D_2, G_2)$, where $G_1$ and $G_2$ are the generators and $D_1$ and $D_2$ are the discriminators. The hypothesis $\mathcal{H}_1$ is that $M_1$ is better than $M_2$ if $G_1$ fools $D_2$ more than $G_2$ fools $D_1$, and vice versa for the hypothesis $\mathcal{H}_0$. The likelihood-ratio is defined as:

$$\frac{p(x|y=1; M_1')}{p(x|y=1; M_2')} = \frac{p(y=1|x; D_1)p(x; G_2)}{p(y=1|x; D_2)p(x; G_1)}, \tag{3}$$

where $M_1'$ and $M_2'$ are the swapped pairs $(D_1, G_2)$ and $(D_2, G_1)$, and $p(x|y=1, M)$ is the likelihood of $x$ generated from the data distribution $p(x)$ by model $M$ and $p(y=1|x; D)$ indicates that discriminator $D$ thinks $x$ is a real sample. To evaluate this, we measure the ratio of how frequently $G_1$, the generator from model 1, fools $D_2$, the discriminator from model 2, and vice-versa: $\frac{D_1(x_2)}{D_2(x_1)}$, where $x_1 \sim G_1$ and $x_2 \sim G_2$. There are two main caveats to the Generative Adversarial Metric. First, the measurement only provides comparisons between pairs of models. Second, the metric has a constraint where the two discriminators must have an approximately similar performance on a calibration dataset, which can be difficult to satisfy in practice.

The **Inception Score** (Salimans et al., 2016) (IS) measures the performance of a model using a third-party neural network trained on a supervised classification task, e.g. Imagenet. The IS computes the expectation of divergence between the distribution of class predictions for samples from the GAN compared to the distribution of class to the distribution of class labels used to train the third-party network,

$$\exp\left(\mathbb{E}_{x \sim Q_\theta} KL(p(y|x)\|p(y))\right). \tag{4}$$

Here, the class prediction given a sample $x$ is computed using the third-party neural network. In (Salimans et al., 2016), Google's Inception Network (Szegedy et al., 2015) trained on Imagenet was the third-party neural network. IS is the most widely used metric to measure GAN performance. However, summarizing samples as the class prediction from a network trained for a different task discards much of the important information in the sample. In addition, it requires another neural network that is trained separately via supervised learning. We demonstrate an example of a failure case of IS in the Experiments section.

The **Fréchet Inception Distance** (FID) (Heusel et al., 2017) extends upon IS. Instead of using the final classification outputs from the third-party network as representations of samples, it uses a representation computed from a late layer of the third-party network. It compares the mean $m_Q$ and covariance $C_Q$ of the Inception-based representation of samples generated by the GAN to the mean $m_P$ and covariance $C_P$ of the same representation for training samples:

$$D^2\left((m_P, C_P), (m_Q, C_Q)\right) = \|m_P - m_Q\|_2^2 + \text{Tr}\left(C_P + C_Q - 2(C_P C_Q)^{\frac{1}{2}}\right), \tag{5}$$

This method relies on the Inception-based representation of the samples capturing all important information and the first two moments of the distributions being descriptive of the distribution.

**Classifier Two-Sample Tests** (C2ST) (Lopez-Paz & Oquab, 2016) proposes training a classifier, similar to a discriminator, that can distinguish real samples from $P$ from generated samples from $Q$, and using the error rate of this classifier as a measure of GAN performance. In their work, they used single-layer and $k$-nearest neighbor (KNN) classifiers trained on a representation of the samples computed from a late layer of a third-party network (in this case, ResNet (He et al., 2015)). C2ST is an IPM (Sriperumbudur et al., 2009), like the MMD and Wasserstein metrics we propose, with $\mu(f) = f$ and $\upsilon(f) = f$, but with a different function class $\mathcal{F}$, corresponding to the family of classifiers chosen (in this case, single-layer networks or KNN, see see our detailed explanation in Appendix 5). The accuracy of a classifier trained to distinguish samples from distributions $P$ and $Q$ is just one way to measure the distance between these distributions, and, in this work, we propose a general family.

# 3 EVALUATION METRICS

Given a generator $G_\theta$ with parameters $\theta$ which generates samples from the distribution $Q_\theta$, we propose to measure the quality of $G_\theta$ by estimating divergence between the true data distribution $P$ and $Q_\theta$ for different choices of divergence measure. We train both $G_\theta$ and $D_\varphi$ on a training data set, and measure performance on a separate test set. See Algorithm 1 for details. We consider metrics from two widely studied divergence and distance measures, $f$-divergence (Nguyen et al., 2008) and the Integral Probability Metric (IPM) (Muller, 1997). In our experiments, we consider the following four metrics that are commonly used to train GANs. Below, $\varphi$ represents the parameters of the discriminator network and $\theta$ represents the parameters of the generator network.

*Original GAN Criterion (GC)*

Training a standard GAN corresponds to minimizing the following (Goodfellow et al., 2014):

$$\max_{\varphi} \quad \mathbb{E}_{x \sim p(x)}[\log(D_\varphi(x))] + \mathbb{E}_{z \sim p(z)}[\log(1 - D_\varphi(G_\theta(z)))], \tag{6}$$

where $p(z)$ is the prior distribution of the generative network and $G_\theta(z)$ is a differentiable function from $z$ to the data space represented by a neural network with parameter $\theta$. $D_\varphi$ is trained with a sigmoid activation function, thus its output is guaranteed to be positive.

*Least-Squares GAN Criterion (LS)*

A Least-Squares GAN corresponds to training with a Pearson $\chi^2$ divergence (Mao et al., 2017):

$$\max_{\varphi} \quad -\mathbb{E}_{x \sim p(x)}[(D_\varphi(x) - b)^2] - \mathbb{E}_{z \sim p(z)}[(D_\varphi(G_\theta(z) - a))^2]. \tag{7}$$

Following (Mao et al., 2017), we set $a = 0$ and $b = 1$ when training $D_\varphi$.

*Maximum Mean Discrepancy (MMD)* The maximum mean discrepancy metric considers the largest difference in the expectations over a unit ball of RKHS $\mathcal{H}$,

$$\max_{\varphi:\|f\|_\mathcal{H} \leq 1} \mathbb{E}_{x \sim p(x)}[D_\varphi(x)] - \mathbb{E}_{z \sim p(z)}[D_\varphi(G_\theta(z)] \tag{8}$$

$$= \mathbb{E}_{x,x' \sim P}\left[k(x,x')\right] + \mathbb{E}_{z,z' \sim p(Z)}\left[k(G_\theta(z), G_\theta(z'))\right] - 2\mathbb{E}_{x \sim P, z \sim p(Z)}\left[k(x, G_\theta(z))\right], \tag{9}$$

where $\mathcal{H}$ is the RKHS with kernel $k(\cdot, \cdot)$ (Gretton et al., 2012). In this case, we do not need to train a discriminator $D_\varphi$ to evaluate our metric.

*Improved Wasserstein Distance (IW)*

Arjovsky & Bottou (2017) proposed the use of the dual representation of the Wasserstein distance (Villani, 2009) for training GANs. The Wasserstein distance is an IPM which considers the 1-Lipschitz function class $\varphi : \|\varphi\|_L \leq 1$:

$$\max_{\varphi:\|\varphi\|_L \leq 1} \left[\mathbb{E}_{x \sim p(X)}\left[D_\varphi(x)\right] - \mathbb{E}_{z \sim p(Z)}\left[D_\varphi(G_\theta(z))\right]\right]. \tag{10}$$

Note that IW (Danihelka et al., 2017) and MMD (Sutherland et al., 2017) were recently proposed to evaluate GANs, but have not been compared before.

---

**Algorithm 1** Compute the divergence/distance.

---

1: **procedure** DIVERGENCECOMPUTATION(Dataset $\{X_{tr}, X_{te}\}$, generator $G_\theta$, learning rate $\eta$, evaluation criterion $J(\varphi, X, Y)$)
2:     Initialize critic network parameter $\varphi$.
3:     **for** $i = 1 \cdots N$ **do**
4:         Sample data points from X, $\{x_m\} \sim X_{tr}$.
5:         Sample points from generative model, $\{s_m\} \sim G_\theta$.
6:         $\varphi \leftarrow \varphi + \eta\nabla_\varphi J(\{x_m\}, \{s_m\}; \varphi)$.
7:     Sample points from generative model, $\{s_m\} \sim G_\theta$.
8:     return $J(\varphi, X_{te}, \{s_m\})$.

---

## 4 EXPERIMENTS

The goals in our experiments are two-fold. First, we wanted to evaluate the metrics we proposed for evaluating GANs. Second, we wanted to use these metrics to evaluate GAN frameworks and architectures. In particular, we evaluated how size of the discriminator and generator networks affected performance, and the sensitivity of each algorithm to training data set size.

*GAN frameworks*. We conducted our experiments on three types of GANs: Deep Convolutional Generative Adversarial Networks (DCGAN), Least-Squares GANs (LS-DCGAN), and Wasserstein GANs (W-DCGAN). Note that to not confuse the test metric names with the GAN frameworks we evaluated, we use different abbreviations. GC is the original GAN criterion, which is used to train DCGANs. The LS criterion is used to train the LS-DCGAN, and the IW is used to train the W-DCGAN.

*Evaluation criteria*. We evaluated these three families of GANs with six metrics. We compared our four proposed metrics to the two most commonly used metrics for evaluating GANs, the IS and FID. Because the optimization of a discriminator is required both during training and test time, we will call the discriminator learned for evaluaton of our metrics the *critic*, in order to not confuse the two discriminators.

We also compared these metrics to human perception, and had three volunteers evaluate and compare sets of images, either from the training data set or generated from different GAN frameworks during training.

*Data sets*. In our experiments, we considered the MNIST (LeCun et al., 1998), CIFAR10, LSUN Bedroom, and Fashion MNIST datasets. MNIST consists of 60,000 training and 10,000 test images with a size of $28 \times 28$ pixels, containing handwritten digits from the classes 0 to 9. From the 60,000 training examples, we set aside 10,000 as validation examples to tune various hyper-parameters. Similarly, FashionMNIST consists exactly the same number of training and test examples. Each example is a 28x28 grayscale image, associated with a label from 10 classes. The CIFAR10 dataset [1] consists of images with a size of $32 \times 32 \times 3$ pixels, with ten different classes of objects. We used 45,000, 5,000, and 10,000 examples as training, validation, and test data, respectively. The LSUN Bedroom dataset consists of images with a size of $64 \times 64$ pixels, depicting various bedrooms. From the 3,033,342 images, we used 90,000 images as training data and 90,000 images as validation data. The learning rate was selected from discrete ranges and chosen based on a held-out validation set.

*Hyperparameters*. Table 10 in the Appendix shows the learning rates and the convolutional kernel sizes that were used for each experiment. The architecture of each network is presented in the Appendix in Figure 10. Additionally, we used exponential-mean-square kernels with several different sigma values for MMD. A pre-trained logistic regression and pre-trained residual network were used for IS and FID on the MNIST and CIFAR10 datasets, respectively. For every experiment, we retrained 10 times with different random seeds, and report the mean and standard deviation.

### 4.1 QUALITATIVE OBSERVATIONS ABOUT EXISTING METRICS

The log-likelihood measurement is the most commonly used metric for generative models. We measured the log-likelihood using AIS[2] on GANs is strange, as shown in Figure 1. We measured the log-likelihood of the DCGAN on MNIST with three different variances, $\sigma^2 = 0.01, 0.025$, and $0.05$. The figure illustrates that the log-likelihood curve over the training epochs varies substantially depending on the variance, which indicates that the fixed Gaussian observable model might not be the ideal assumption for GANs. Moreover, we observe a high log-likelihood at the beginning of training, followed by a drop in likelihood, which then returns to the high value.

The IS and MMD metrics do not require training a critic. It was easy to find samples for which IS and MMD scores did not match their visual quality. For example, Figure 2 shows samples generated by a DCGAN when it failed to train properly. Even though the failed DCGAN samples are much darker than the samples on the right, the IS for the left samples is higher/better than for the right samples.

---

[1] https://github.com/Lasagne/Recipes/blob/master/papers/deep_residual_learning/Deep_Residual_Learning_CIFAR10.py

[2] We used the original source code from https://github.com/tonywu95/eval_gen

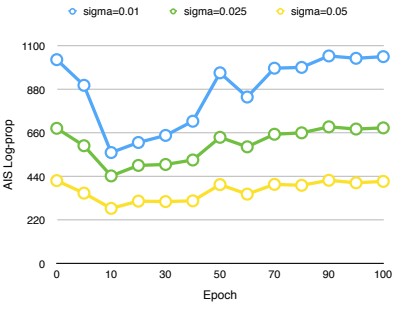

Figure 1: Log-likelihood estimated using AIS for generators learned using DCGAN at various points during training, MNIST data set.

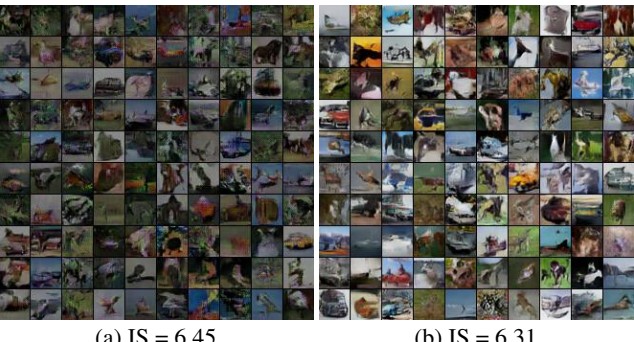

(a) IS = 6.45                    (b) IS = 6.31

Figure 2: Misleading examples of Inception Scores.

As the Imagenet-trained network is likely trained to be somewhat invariant to overall intensity, this issue is to be expected.

A failure case for MMD is shown in Figure 5. The samples on the right are dark, like the previous examples, but still textually recognizable, whereas the samples on the left are totally meaningless. However, MMD gives lower/worse distances to the left samples. The average intensity of the pixels of the left samples are closer to that for the training data, suggesting that MMD is overly sensitive to image intensity. Thus, IS is under-sensitive to image intensity, while MMD if oversensitive to it. In Section 4.2.1, we conduct more systematic experiments by measuring the correlation between these metrics to human perceptual scores.

## 4.2 METRIC COMPARISON

To both compare the metrics as well as different GAN frameworks, we evaluated the six metrics on different GAN frameworks. Tables 2, 3, and 4 present the results on MNIST, CIFAR10, and LSUN respectively.

As each type of GAN was trained using one of our proposed metrics, we investigated whether the metric favors samples from the model trained using the same metric. Interestingly, we do not see this behavior, and our proposed metrics agree on which GAN framework produces samples closest to the test data set. Every metric, except for MMD, showed that LS-DCGAN performed best for MNIST and CIFAR10, while W-DCGAN performed best for LSUN. As discussed below, we found DCGAN to be unstable to train, and thus excluded GC as a metric for experiments except for this first data set. For Fashion-MNIST, FID's ranking disagreed with IW and LS.

We observed similar results for a range of different critic CNN architectures (number of feature maps in each convolutional layer): $[3, 64, 128, 256]$, $[3, 128, 256, 512]$, $[3, 256, 512, 1024]$, and $[3, 320, 640, 1280]$ (see Supp. Fig. 12 and 13).

We evaluated a larger variety of GAN frameworks using pre-trained GANs downloaded from (pyt). In particular, we evaluated on EBGAN(Junbo Zhao, 2016), BEGAN(Berthelot et al., 2017), W-DCGAN GP(Gulrajani et al., 2017), and DRAGAN (Kodali et al., 2017). Table 5 presents the evaluation results. Critic architectures were selected to match those of these pre-trained GANs. For both MNIST and FashionMNIST, the three metrics are consistent and they rank DRAGAN the highest, followed by LS-DCGAN and DCGAN.

The standard deviations for the IW distance are higher than for LS divergence. We computed the Wilcoxon rank sum in order to test that whether medians of the distributions of distances are the same for DCGAN, LS-DCGAN, and W-DCGAN. We found that the different GAN frameworks have significantly different performance according to the LS-GAN criterion, but not according to the IW criterion ($p < .05$, Wilcoxon rank-sum test). Thus LS is more sensitive than IW.

We evaluated the consistency of the metrics with respect to the size of validation set. We trained our three GAN frameworks for 100 epochs with training 90,000 examples from the LSUN Bedroom

Table 2: GAN scores for various metrics trained on MNIST. Lower values are better for MMD, IW, LS, GC, and FID, higher values are better for IS. Lighter color indicates better performance.

| Model | MMD | IW | GC | LS | IS (Logistic Reg.) |
|---|---|---|---|---|---|
| DCGAN | $0.028 \pm 0.0066$ | $7.01 \pm 1.63$ | $-2.2e-3 \pm 3e-4$ | $-0.12 \pm 0.013$ | $5.76 \pm 0.10$ |
| W-DCGAN | $0.006 \pm 0.0009$ | $7.71 \pm 1.89$ | $-4e-4 \pm 4e-4$ | $-0.05 \pm 0.008$ | $5.17 \pm 0.11$ |
| LS-DCGAN | $0.012 \pm 0.0036$ | $4.50 \pm 1.94$ | $-3e-3 \pm 6e-4$ | $-0.13 \pm 0.022$ | $6.07 \pm 0.08$ |

Table 3: GAN scores for various metrics trained on CIFAR10.

| Model | MMD | IW | LS | IS (ResNet) | FID |
|---|---|---|---|---|---|
| DCGAN | $0.0538 \pm 0.014$ | $8.844 \pm 2.87$ | $-0.0408 \pm 0.0039$ | $6.649 \pm 0.068$ | $0.112 \pm 0.010$ |
| W-DCGAN | $0.0060 \pm 0.001$ | $9.875 \pm 3.42$ | $-0.0421 \pm 0.0054$ | $6.524 \pm 0.078$ | $0.095 \pm 0.003$ |
| LS-DCGAN | $0.0072 \pm 0.0024$ | $7.10 \pm 2.05$ | $-0.0535 \pm 0.0031$ | $6.761 \pm 0.069$ | $0.088 \pm 0.008$ |

Table 4: GAN scores for various metrics trained on LSUN Bedroom dataset.

| Model | MMD | IW | LS |
|---|---|---|---|
| DCGAN | 0.00708 | 3.79097 | -0.14614 |
| W-DCGAN | 0.00584 | 2.91787 | -0.20572 |
| LS-DCGAN | 0.00973 | 3.36779 | -0.17307 |

Table 5: Evaluation of GANs on MNIST and Fashion-MNIST datasets.

| Model | MNIST | | | Fashion-MNIST | | |
|---|---|---|---|---|---|---|
| | IW | LS | FID | IW | LS | FID |
| DCGAN | $0.4814 \pm 0.0083$ | $-0.111 \pm 0.0074$ | $1.84 \pm 0.15$ | $0.69 \pm 0.0057$ | $-0.0202 \pm 0.00242$ | $3.23 \pm 0.34$ |
| EBGAN | $0.7277 \pm 0.0159$ | $-0.029 \pm 0.0026$ | $5.36 \pm 0.32$ | $0.99 \pm 0.0001$ | $-2.2e-5 \pm 5.3e-5$ | $104.08 \pm 0.56$ |
| W-DCGAN GP | $0.7314 \pm 0.0194$ | $-0.035 \pm 0.0059$ | $2.67 \pm 0.15$ | $0.89 \pm 0.0086$ | $-0.0005 \pm 0.00037$ | $2.56 \pm 0.25$ |
| LS-DCGAN | $0.5058 \pm 0.0117$ | $-0.115 \pm 0.0070$ | $2.20 \pm 0.27$ | $0.68 \pm 0.0086$ | $-0.0208 \pm 0.00290$ | $0.62 \pm 0.13$ |
| BEGAN | - | $-0.009 \pm 0.0063$ | $15.9 \pm 0.48$ | $0.90 \pm 0.0159$ | $-0.0016 \pm 0.00047$ | $1.51 \pm 0.16$ |
| DRAGAN | $0.4632 \pm 0.0247$ | $-0.116 \pm 0.0116$ | $1.09 \pm 0.13$ | $0.66 \pm 0.0108$ | $-0.0219 \pm 0.00232$ | $0.97 \pm 0.14$ |

dataset. We then trained LS and IW critics using both 300 and 90,000 validation examples. We looked at how often the critic trained with 300 examples agreed with that trained with 90,000 examples. The LS critics agreed 88% of the time, while the IW critics agreed only 55% of the time (slightly better than chance). Thus, LS is more robust to validation data set size. Another advantage is that measuring the LS distance is faster than measuring the IW distance, as estimating IW involves regularizing with a gradient penalty time (Gulrajani et al., 2017). Computing the gradient penalty term and tuning its regularization coefficient requires extra computational time.

As mentioned above, we found training a critic using the GC criterion (corresponding to a DCGAN) to be unstable. It has previously been speculated that this is the case because the support of the data and model distributions possibly becoming disjoint (Arjovsky & Bottou, 2017), and the Hessian of the GAN objective being non-Hermitian (Mescheder et al., 2017). LS-DCGAN and W-DCGAN are proposed to address this by providing non-saturating gradients. We also found DCGAN to be difficult to train, and thus only report results using the corresponding criterion GC for MNIST. Note that this is different than training a discriminator as part of standard GAN training because we are training from a random initialization, not from the previous version of the discriminator.

Our experience was that the LS-DCGAN was the simplest and most stable model to train. We visualized the 2D subspace of the loss surface of the GANs in Supp. Fig. 29. Here, we took the parameters of three trained models (corresponds to red vertices in the figure) and applied barycentric interpolation with respect to three parameters (see details from (Im et al., 2016c)). DCGAN surfaces have much sharper slopes when compared to the LS-DCGAN and W-DCGAN, and LS-DCGAN has the most gentle surfaces. In what follows, we show that this geometric view is consistent with our finding that LS-DCGAN is the easiest and the most stable to train.

Table 6: The fraction of pairs of which each metric agrees with human scores. We use colored asterisks to represent significant differences (two-sided Fisher's test, p < .05). E.g. * in the IW row indicates that IW and IS are significantly different.

| Metric | Fraction | [Agreed/Total] samples | p < .05? |
|--------|----------|------------------------|----------|
| IW | 0.977 | 128 / 131 | * * |
| LS | 0.931 | 122 / 131 | * |
| IS | 0.863 | 113 / 131 | * |
| MMD | 0.832 | 109 / 131 | * * |

### 4.2.1 COMPARISON TO HUMAN PERCEPTION

We compared the LS, IW, MMD, and IS metrics to human perception for the CIFAR10 dataset. To accomplish this, we asked five volunteers to choose which of two sets of 100 samples, each generated using a different generator, looked most realistic. Before surveying, the volunteers were trained to choose between real samples from CIFAR10 and samples generated by a GAN. Supp. Fig. 14 displays the user interface for the participants, and Supp. Fig. 15 shows the fraction of labels that the volunteers agreed upon.

Table 6) presents the fraction of pairs for which each metric agrees with humans (higher is better). IW has a slight edge over LS, and both outperform IS and MMD. In Figure 3, we show examples in which all humans agree and metrics disagrees with human perception. All such examples are shown in Supp. Fig. 21-24.

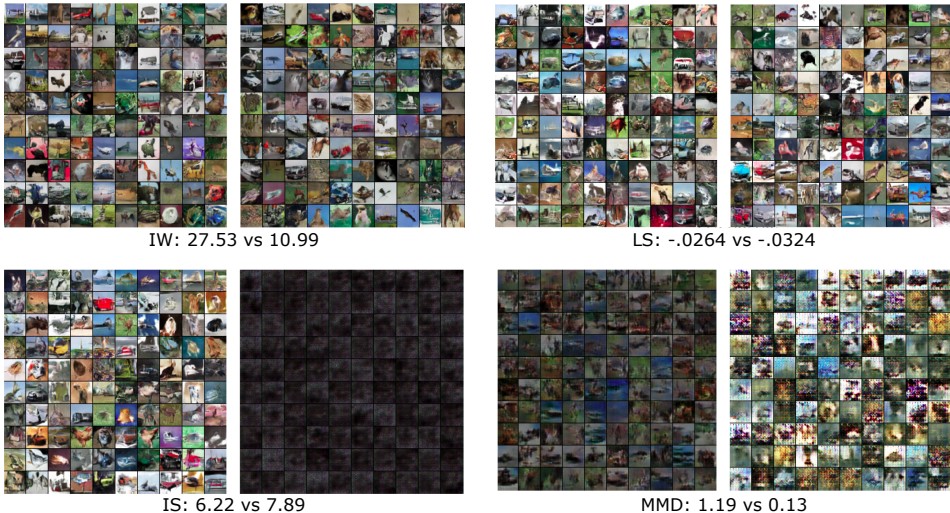

Figure 3: Pairs of generated image sets for which human perception and metrics disagree. Here, we selected one such example for each metric for which the difference in that metric's scores was high. For each pair, humans perceived the set of images on the left to be more realistic than those on the right, while the metric predicted the opposite. Below each pair of images, we indicate the metric's score for the left and right image sets.

### 4.3 SENSITIVITY ANALYSIS

### 4.3.1 PERFORMANCE CHANGE WITH RESPECT TO THE SIZE OF THE NETWORK

Several works have demonstrated an improvement in performance by enlarging deep network architectures (Krizhevsky et al., 2012; Simonyan & Zisserman, 2014; He et al., 2015; Huang et al., 2017). Here, we investigate performance changes with respect to the width and depth of the networks.

First, we trained three GANs with varying numbers of feature map sizes, as shown in Table 7 (a-d). Note that we double the number of feature maps in Table 7 for both the discriminators and generators.

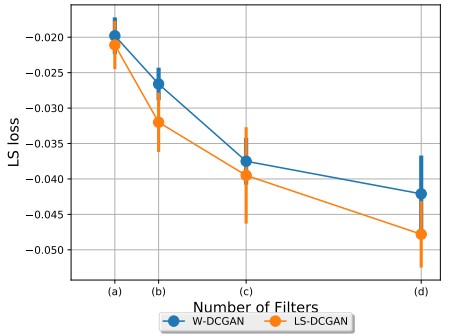

Figure 4: LS score evaluation of W-DCGAN & LS-DCGAN w.r.t number of feature maps.

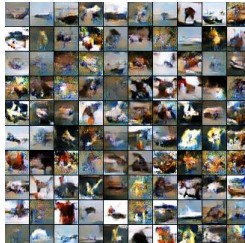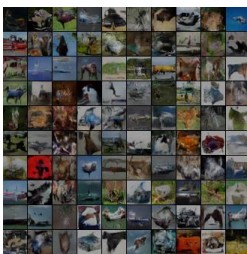

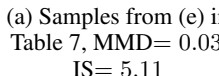

(a) Samples from (e) in Table 7, MMD= 0.03, IS= 5.11

(b) Samples from (f) in Table 7, MMD= 0.49, IS= 6.15

Figure 5: W-DCGAN trained with different numbers of feature maps.

Table 8: LS-DCGAN and W-DCGAN scores on CIFAR10 with respect to different generator and discriminator capacity.

| Model | Architecture (Table 7) | MMD Test vs. Samples | IW | LS | IS (ResNet) |
|---|---|---|---|---|---|
| W-DCGAN | (e) | $0.1057 \pm 0.0798$ | $450.17 \pm 25.74$ | $-0.0079 \pm 0.0009$ | $6.403 \pm 0.839$ |
| | (f) | $0.2176 \pm 0.2706$ | $16.52 \pm 15.63$ | $-0.0636 \pm 0.0101$ | $6.266 \pm 0.055$ |
| LS-DCGAN | (e) | $0.1390 \pm 0.1525$ | $343.23 \pm 47.55$ | $-0.0092 \pm 0.0007$ | $5.751 \pm 0.511$ |
| | (f) | $0.0054 \pm 0.0022$ | $12.75 \pm 4.29$ | $-0.0372 \pm 0.0068$ | $6.600 \pm 0.061$ |

In Figure 4, the performance of the LS score increases logarithmically as the number of feature maps is doubled. A similar behaviour is observed in other metrics as well (see S.M. Figure 16).

We then analyzed the importance of size in the discriminative and generative networks. We considered two extreme feature map sizes, where we choose a small and large number of feature maps for the generator and discriminator, and vice versa (see label (e) and (f) in Table 7), and results are shown in Table 8. For LS-DCGAN, it can be seen that a large number of feature maps for the discriminator has a better score than a large number of feature maps for the generator. This can also be qualitatively verified by looking at the samples from architectures (a), (e), (f), and (d) in Figure 6. For W-DCGAN, we observe the agreement between the LS and IW metric

Table 7: Reference for the different architectures explored in the experiments.

| Label | Feature Maps | |
|---|---|---|
| | Discriminator | Generator |
| (a) | [3, 16 , 32 , 64 ] | [128 , 64 , 32 , 3] |
| (b) | [3, 32 , 64 , 128] | [256 , 128, 64 , 3] |
| (c) | [3, 64 , 128, 256] | [512 , 256, 128, 3] |
| (d) | [3, 128, 256, 512] | [1024, 512, 256, 3] |
| (e) | [3, 16 , 32 , 64 ] | [1024, 512, 256, 3] |
| (f) | [3, 128, 256, 512] | [128 , 64 , 32 , 3] |

and conflict with MMD and IS. When we look at the samples from the W-DCGAN in Figure 5, it is clear that the model with a larger number of feature maps in the discriminator should get a better score; this is another example of false intuition propagated by MMD and IS. One interesting observation is that when we compare the score and samples from architecture (a) and (e) from Table 7, architecture (a) is much better than (e) (see Figure 6). This demonstrates that having a large generator and small discriminator is worse than having a small architecture for both networks. Overall, we found that having a larger generator than discriminator does not give good results, and that it is more desirable to have a larger discriminator than generator. Similar results were also observed for MNIST, as shown in S.M. Figure 20. This result somewhat supports the theoretical result from Arora et al. (2017), where the generator capacity needs to be modulated in order for approximately pure equilibrium to exist for GANs.

Lastly, we experimented with how performance changes with respect to the dimension of the noise vectors. The source of the sample starts by transforming a noise vector into a meaningful image. It is unclear how the size of noise affects the ability of the generator to generate a meaningful image. Che et al. (2017) have observed that a 100-d noise vector preserves modes better than a 200-d noise vector

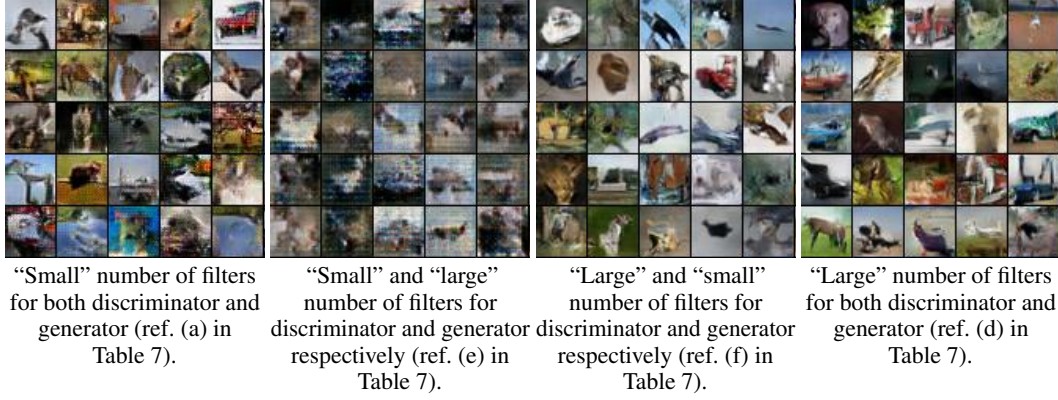

| "Small" number of filters for both discriminator and generator (ref. (a) in Table 7). | "Small" and "large" number of filters for discriminator and generator respectively (ref. (e) in Table 7). | "Large" and "small" number of filters for discriminator and generator respectively (ref. (f) in Table 7). | "Large" number of filters for both discriminator and generator (ref. (d) in Table 7). |
|---|---|---|---|

Figure 6: Samples from different LS-DCGAN architectures.

for DCGAN. Our experiments show that this depends on the model. Given a fixed size architecture (d) from Table 7, we observed the performance of LS-DCGAN and W-DCGAN by varying the size of noise vector $z$. Table 9 illustrates that LS-DCGAN gives the best score with a noise dimension of 50 and W-DCGAN gives best score with a noise dimension of 150 for both IW and LS. The outcome of LS-DCGAN is consistent with the result in (Che et al., 2017). It is possible that this occurs because both models fall into the category of $f$-divergences, whereas the W-DCGAN behaves differently because its metric falls under a different category, the Integral Probability Metric.

Table 9: LS-DCGAN and W-DCGAN scores on CIFAR10 with respect to the dimensionality of the noise vector.

| $|z|$ | LS-DCGAN | | W-DCGAN | |
|---|---|---|---|---|
| | IW | LS | IW | LS |
| 50 | $3.9010 \pm 0.60$ | $-0.0547 \pm 0.0059$ | $6.0948 \pm 3.21$ | $-0.0532 \pm 0.0069$ |
| 100 | $5.6588 \pm 1.47$ | $-0.0511 \pm 0.0065$ | $5.7358 \pm 3.25$ | $-0.0528 \pm 0.0051$ |
| 150 | $5.8350 \pm 0.80$ | $-0.0434 \pm 0.0036$ | $3.6945 \pm 1.33$ | $-0.0521 \pm 0.0050$ |

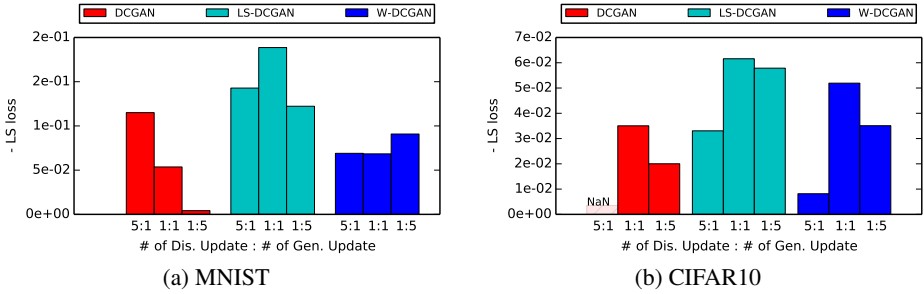

(a) MNIST    (b) CIFAR10

Figure 7: LS score evaluation with respect to a varying number of discriminator and generator updates on DCGAN, W-DCGAN, and LS-DCGAN.

### 4.3.2 PERFORMANCE CHANGE WITH RESPECT TO THE RATIO OF NUMBER OF UPDATES BETWEEN THE GENERATOR AND DISCRIMINATOR

In practice, we alternate between updating the discriminator and generator, and yet this is not guaranteed to give the same result as the solution to the min-max problem in Equation 2. Hence, the update ratio can influence the performance of GANs. We experimented with three different update ratios, $5:1$, $1:1$, and $1:5$, with respect to the discriminator and generator update. We applied these ratios to both the MNIST and CIFAR10 datasets on all models.

Figure 7 presents the LS scores on both MNIST and CIFAR10 and this result is consistent with the IW metric as well (see S.M. Figure 25). However, we did not find that any one update ratio was superior over others between the two datasets. For CIFAR10, the $1:1$ update ratio worked best for all models, and for MNIST, different ratios worked better for different models. Hence, we conclude that number of update ratios for each model needs to be dynamically tuned. The corresponding samples from the models trained by different update ratios are shown in S.M. Figure 27.

### 4.3.3 PERFORMANCE WITH RESPECT TO THE AMOUNT OF AVAILABLE TRAINING DATA

In practice, DCGANs are known to be unstable, and the generator tends to suffer as the discriminator gets better due to disjoint support between the data and generator distributions (Goodfellow et al., 2014; Arjovsky & Bottou, 2017). W-DCGAN and LS-DCGAN offer alternative ways to solving this problem. If the model is suffering from disjoint support, having more training examples will not help, and alternatively, if the model does not suffer from such a problem, having more training examples could potentially help.

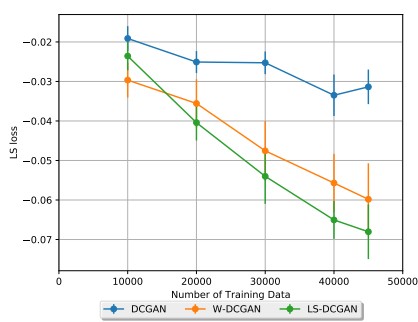

Figure 8: LS score evaluation on W-DCGAN & LS-DCGAN w.r.t number of data points.

Here, we explore the sensitivity of three different kinds of GANs with respect to the number of training examples. We have trained GANs with 10,000, 20,000, 30,000, 40,000, and 45,000 examples on CIFAR10. Figure 8 shows that the LS score curve of DCGAN grows quite slowly when compared to W-DCGAN and LS-DCGAN. The three GANs have a relatively similar loss when they are trained with 10,000 training examples. However, the DCGAN only gained $0.0124 \pm 0.00127$ by increasing from 10,000 to 40,000 training examples, whereas the performance of W-DCGAN and LS-DCGAN improved by $0.03016 \pm 0.00469$ and $0.0444 \pm 0.0033$, respectively. Thus, we empirically observe that W-DCGAN and LS-DCGAN have faster performance increases than a DCGAN as the number of training examples grows.

## 5 CONCLUSION

In this paper, we proposed to use four well-known distance functions as an evaluation metrics, and empirically investigated the DCGAN, W-DCGAN, and LS-DCGAN families under these metrics. Previously, these models were compared based on visual assessment of sample quality and difficulty of training. In our experiments, we showed that there are performance differences in terms of average experiments, but that some are not statistically significant. Moreover, we thoroughly analyzed the performance of GANs under different hyper-parameter settings.

There are still several types of GANs that need to be evaluated, such as GRAN (Im et al., 2016a), IW-DCGAN (Gulrajani et al., 2017), BEGAN (Berthelot et al., 2017), MMDGAN (Li et al., 2017), and CramerGAN (Bellemare et al., 2017). We hope to evaluate all of these models under this framework and thoroughly analyze them in the future. Moreover, there has been an investigation into taking ensemble approaches to GANs, such as Generative Adversarial Parallelization Im et al. (2016b). Ensemble approaches have been empirically shown to work well in many domains of research, so it would be interesting to find out whether ensembles can also help in min-max problems. Alternatively, we can also try to evaluate other log-likelihood-based models like NVIL (Mnih & Gregor, 2014), VAE (Kingma & Welling, 2014), DVAE (Im et al., 2015), DRAW (Gregor et al., 2015), RBMs (Hinton et al., 2006; Salakhutdinov & Hinton, 2009), NICE Dinh et al. (2014), etc.

Model evaluation is an important and complex topic. Model selection, model design, and even research direction can change depending on the evaluation metric. Thus, we need to continuously explore different metrics and rigorously evaluate new models.

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

## APPENDIX

### RELATIONSHIP BETWEEN METRICS AND BINARY CLASSIFICATION

In this paper, we considered four distance metrics that belong to two class of metrics, $\phi$-divergence and IPMs. Sriperumbudur et al. (2009) have shown that the optimal risk function is associated with a binary classifier with $P$ and $Q$ distributions conditioned on a class when the discriminant function is restricted to certain $F$ (Theorem 17 from (Sriperumbudur et al., 2009)).

Let the optimal risk function be:

$$R(L, F) = \inf_{f \in F} \int L(y, f(x)) dp(x, y), \tag{11}$$

where $F$ is the set of discriminant functions (classifier), $y \in -1, 1$, and $L$ is the loss function.

By following derivation, we can see that the optimal risk function becomes IPM:

$$R(L, F) = \inf_{f \in F} \int L(y, f(x)) du(x, y) \tag{12}$$

$$= \inf_{f \in F} \left[ \epsilon \int L(1, f(x)) dp(x) + (1 - \epsilon) \int L(0, f(x)) dq(x) \right] \tag{13}$$

$$= \inf_{f \in F} f dp(x) + \inf_{f \in F} f dq(x) \tag{14}$$

$$= -IPM \tag{15}$$

where $L(1, f(x)) = 1/\epsilon$ and $L(0, f(x)) = -1/(1 - \epsilon)$.

The second equality is derived by separating the loss for class 1 and class 0. The third equality is from the way how we chose L(1,f(x)) and L(0,f(x)). The last equality is derived from that fact that $F$ is symmetric around zero ($f \in F => -f \in F$). Hence, this shows that with appropriately choosing $L$, MMD and Wasserstein distance can be understood as the optimal $L$-risk associated with binary classifier with specific set of $F$ functions. For example, Wasserstein distance and MMD distances are equivalent to the optimal risk function with 1-Lipschitz classifiers and a RKHS classifier with an unit length.

### EXPERIMENTAL HYPER-PARAMETERS

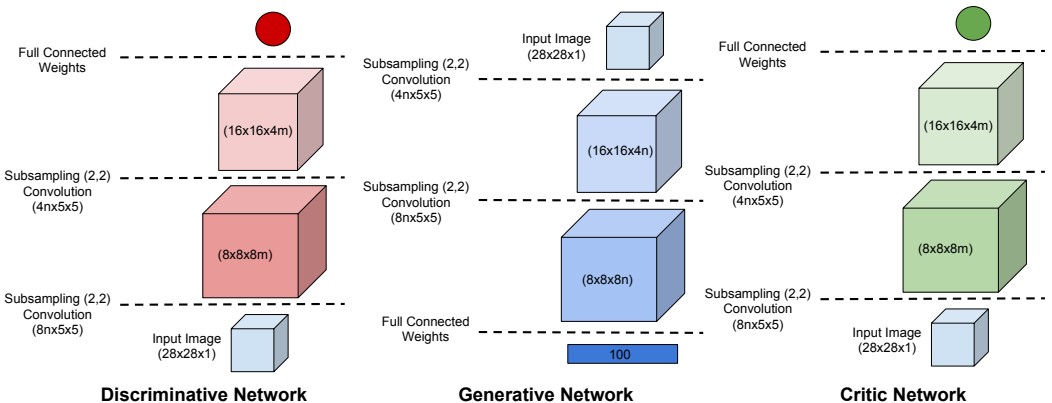

Figure 9: GAN Topology for MNIST.

Figure 10: GAN Topology for CIFAR10.

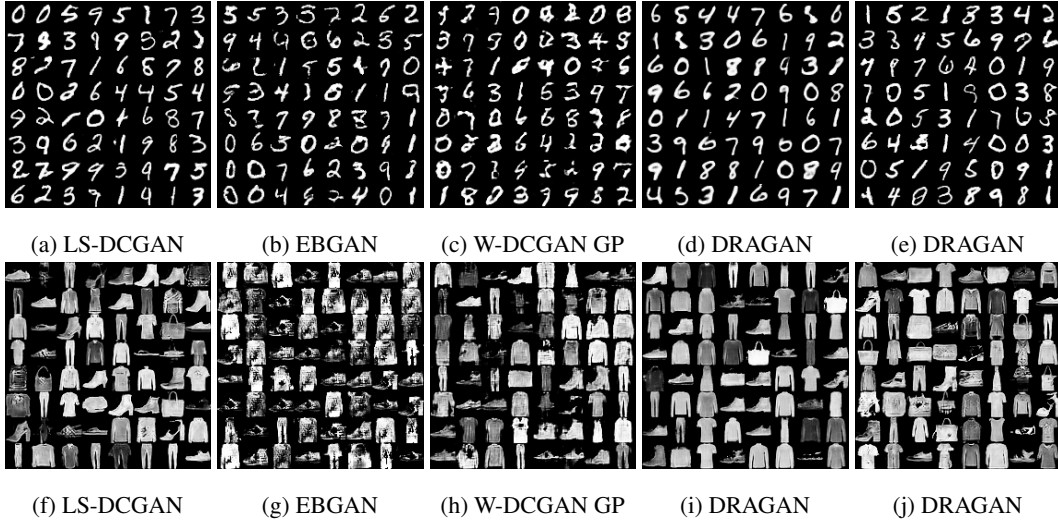

(a) LS-DCGAN  (b) EBGAN  (c) W-DCGAN GP  (d) DRAGAN  (e) DRAGAN

(f) LS-DCGAN  (g) EBGAN  (h) W-DCGAN GP  (i) DRAGAN  (j) DRAGAN

Figure 11: MNIST & FashionMNIST Samples

Table 10: Hyper-parameters used for different experiments.

| | | GAN training | | | Critic Training (test time) | | |
|---|---|---|---|---|---|---|---|
| | Model | Disc. Lr. | Gen. Lr. | Ratio[3] | Cr. Lr. | Cr. Kern | Num Epoch |
| Table 2 | DCGAN | 0.0002 | 0.0004 | 1:2 | 0.0001 | [1, 128, 32] | 25 |
| | W-DCGAN | 0.0004 | 0.0008 | 1:1 | | | |
| | LS-DCGAN | 0.0004 | 0.0008 | 1:2 | | | |
| Table 3 | DCGAN | 0.0002 | 0.0001 | 1:2 | 0.0002 | [3, 128, 256, 512] | 11 |
| | W-DCGAN | 0.0008 | 0.0004 | 1:1 | | | |
| | LS-DCGAN | 0.0008 | 0.0004 | 1:2 | | | |
| Table 4 | DCGAN | 0.00005 | 0.0001 | 1:2 | 0.0002 | [3, 128, 256, 512,1024] | 4 |
| | W-DCGAN | 0.0002 | 0.0004 | 1:2 | | | |
| | LS-DCGAN | 0.0002 | 0.0004 | 1:2 | | | |
| Table 5 | ALL GANs | 0.0002 | 0.0002 | 1:1 | 0.0002 | [1, 64, 128] | 25 |
| Table 8 | DCGAN | 0.0002 | 0.0001 | 1:2 | 0.0002 | [3, 128, 256, 512] | 11 |
| | W-DCGAN | 0.0002 | 0.0001 | 1:1 | | | |
| | LS-DCGAN | 0.0008 | 0.0004 | 1:2 | | | |
| Table 11 | ALL GANs | 0.0002 | 0.0002 | 1:1 | 0.0002 | [1, 64, 128] | 25 |
| Table 12 | ALL GANs | 0.0002 | 0.0002 | 1:1 | 0.0002 | [1, 64, 128] | 25 |
| Figure 7b | DCGAN | 0.0001 | 0.00005 | 5:1 | 0.0002 | [3, 128, 256, 512] | 11 |
| | | | | 1:1 | | | |
| | | | | 1:5 | | | |
| | W-DCGAN | | | 5:1 | | | |
| | | | | 1:1 | | | |
| | | 0.0008 | 0.0004 | 1:5 | | | |
| | LS-DCGAN | | | 5:1 | | | |
| | | | | 1:1 | | | |
| | | | | 1:5 | | | |
| Figure 26 | DCGAN | 0.0001 | 0.00005 | 5:1 | 0.0002 | [1, 128, 32] | 25 |
| | | | | 1:1 | | | |
| | | | | 1:5 | | | |
| | W-DCGAN | | | 5:1 | | | |
| | | | | 1:1 | | | |
| | | 0.0008 | 0.0004 | 1:5 | | | |
| | LS-DCGAN | | | 5:1 | | | |
| | | | | 1:1 | | | |
| | | | | 1:5 | | | |
| Figure 16 | DCGAN | 0.0002 | 0.0001 | 1:2 | 0.0002 | [3, 128, 256, 512] | 11 |
| | W-DCGAN | 0.0002 | 0.0001 | 1:1 | | | |
| | LS-DCGAN | 0.0008 | 0.0004 | 1:2 | | | |
| Figure 28 | DCGAN | 0.0002 | 0.0001 | 1:5 | 0.0002 | [3, 256, 512, 1028] | 11 |
| | W-DCGAN | 0.0008 | 0.0004 | 1:1 | | | |
| | LS-DCGAN | 0.0008 | 0.0004 | 1:5 | | | |

MORE EXPERIMENTS

We trained two critics on training data and validation data, respectively, and evaluated on test data from both critics. We trained six GANs (GAN, LS-DCGAN, W-DCGAN GP, DRAGAN, BEGAN, EBGAN) on MNIST and FashionMNIST. We trained these GANs with 50,000 training examples. At test time, we used 10,000 training and 10,000 validation examples for training the critics, and evaluated on 10,000 test examples. Here, we present the test scores from the critics trained on training and validation data. The results are shown in Table **??**. Note that we also have the IW and FID evaluation on these models in the paper. For FashionMNIST, we find that test scores with a critic trained on training and validation data are very close. Hence, we do not see any indication of overfitting. On the other hand, there are gaps between the scores for the MNIST dataset and the test scores from critics trained on the validation set. which gives better performance than the ones that are trained on the training set.

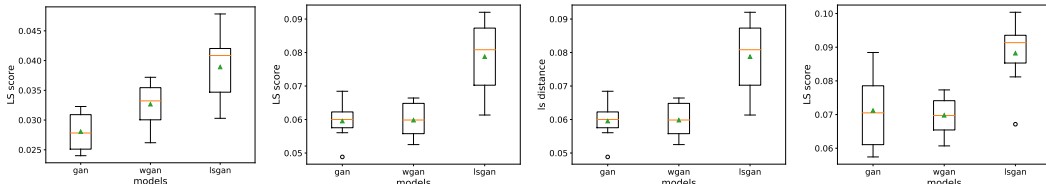

Filter #: [3, 64, 128, 256]   Filter #: [3, 128, 256, 512]  Filter #: [3, 256, 512, 1024]  Filter #: [3, 320, 640, 1280]

Figure 12: GAN evaluation using different architectures for the critic (Number of feature maps in each layer of the CNN critic). Above figures are evaluated under negative least-square loss and Figures 13 are evaluated under Wasserstein distance.

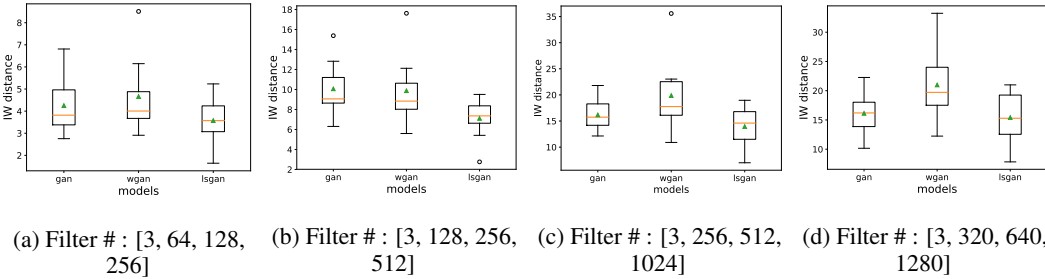

(a) Filter # : [3, 64, 128, 256]    (b) Filter # : [3, 128, 256, 512]    (c) Filter # : [3, 256, 512, 1024]    (d) Filter # : [3, 320, 640, 1280]

Figure 13: GAN evaluation using different critic's architecture (Number of filter of critic's convolutional network). Figure (a,b,c,d) are evaluation under Wasserstein distance.

Table 11: Evaluation of GANs on MNIST dataset. Test score comparison between the two critics that are trained by training and validation dataset.

| Model | LS Score | | IW Score | |
|---|---|---|---|---|
| | Trained on training data | Trained on validation. data | Trained on training data | Trained on validation. data |
| DCGAN | -0.312 ± 0.010 | -0.4408 ± 0.0201 | 0.300 ± 0.0103 | 0.259 ± 0.0083 |
| EBGAN | -3.38e-6 ± 0.1.86e-7 | -3.82e-6 ± 2.82e-7 | 0.999 ± 0.0001 | 0.999 ± 0.0001 |
| WGAN GP | -0.196 ± 0.006 | -0.307 ± 0.0381 | 0.705 ± 0.0202 | 0.635 ± 0.0270 |
| LSGAN | -0.323 ± 0.0104 | -0.352 ± 0.0143 | 0.232 ± 0.0156 | 0.195 ± 0.0103 |
| BEGAN | -0.081 ± 0.016 | -0.140 ± 0.0329 | 0.888 ± 0.0097 | 0.858 ± 0.0131 |
| DRAGAN | -0.318 ± 0.012 | -0.384 ± 0.0139 | 0.266 ± 0.0060 | 0.235 ± 0.0079 |

Table 12: Evaluation of GANs on Fashion-MNIST dataset. Test score comparison between the two critics that are trained by training and validation dataset.

| Model | LS Score | | IW Score | |
|---|---|---|---|---|
| | Trained on training data | Trained on validation. data | Trained on training data | Trained on validation. data |
| DCGAN | -0.1638 ± 0.010 | -0.1635 ± 0.0006 | 0.408 ± 0.0135 | 0.4118 ± 0.0107 |
| EBGAN | -0.0037 ± 0.0009 | -0.0048 ± 0.0023 | 0.415 ± 0.0067 | 0.4247 ± 0.0098 |
| WGAN GP | -0.000175 ± 0.0000876 | -0.000448 ± 0.0000862 | 0.921 ± 0.0061 | 0.9234 ± 0.0059 |
| LSGAN | -0.135 ± 0.0046 | -0.136 ± 0.0074 | 0.631 ± 0.0106 | 0.6236 ± 0.0200 |
| BEGAN | -0.1133 ± 0.042 | -0.0893 ± 0.0095 | 0.429 ± 0.0148 | 0.4293 ± 0.0213 |
| DRAGAN | -0.1638 ± 0.015 | -0.1645 ± 0.0151 | 0.641 ± 0.0304 | 0.6311 ± 0.0547 |

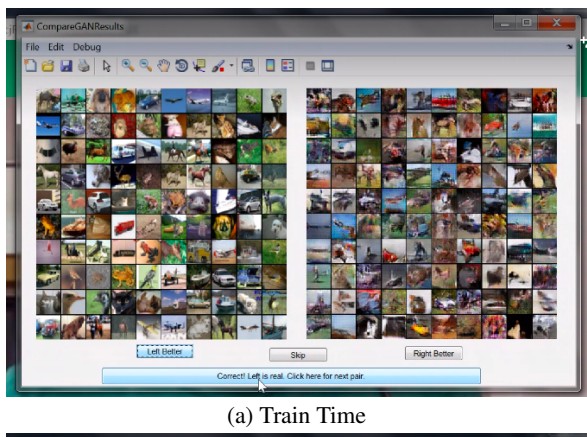

(a) Train Time

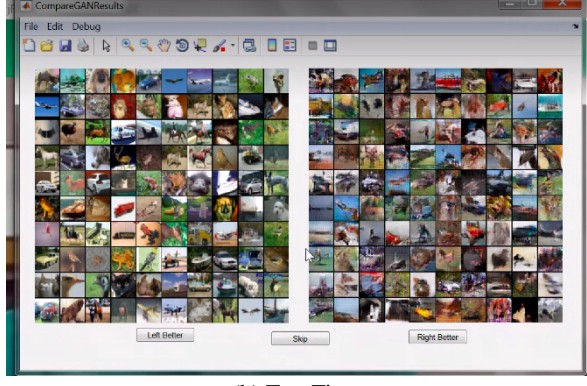

(b) Test Time

Figure 14: The participants are trained by selecting between random samples generated by GANs versus samples from data distribution. They get a positive reward if they selected the data samples and a negative reward if they select the samples from the model. After enough training, they choose the better group of samples among two randomly select set of samples.

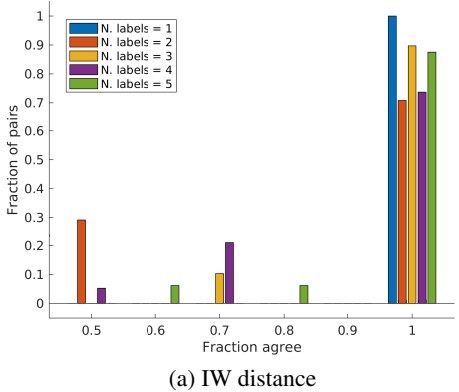

(a) IW distance

Figure 15: The fraction of labels that agree for each pair, depending on the number of labels for each pair, presented as a histogram. By definition, if there is only one participant, that participant must agree with themselves.

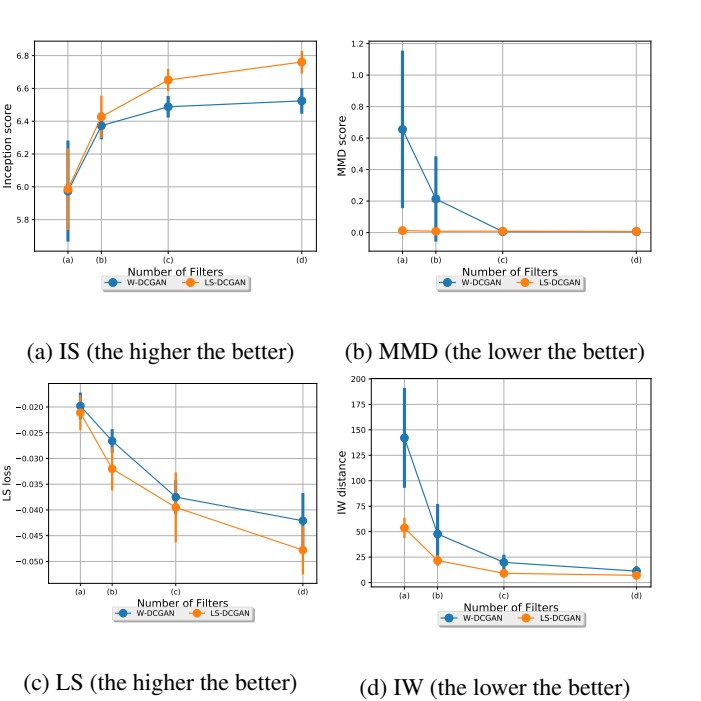

(a) IS (the higher the better)

(b) MMD (the lower the better)

(c) LS (the higher the better)

(d) IW (the lower the better)

Figure 16: Performance of W-DCGAN & LS-DCGAN with respect to number of filters.

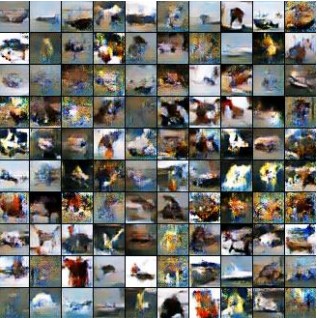

(a) Samples from (e) in Table 7, MMD= 0.03, IS= 5.11

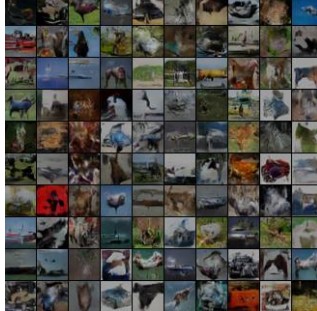

(b) Samples from (f) in Table 7, , MMD= 0.49, IS= 6.15

Figure 17: W-DCGAN trained with different number of filters.

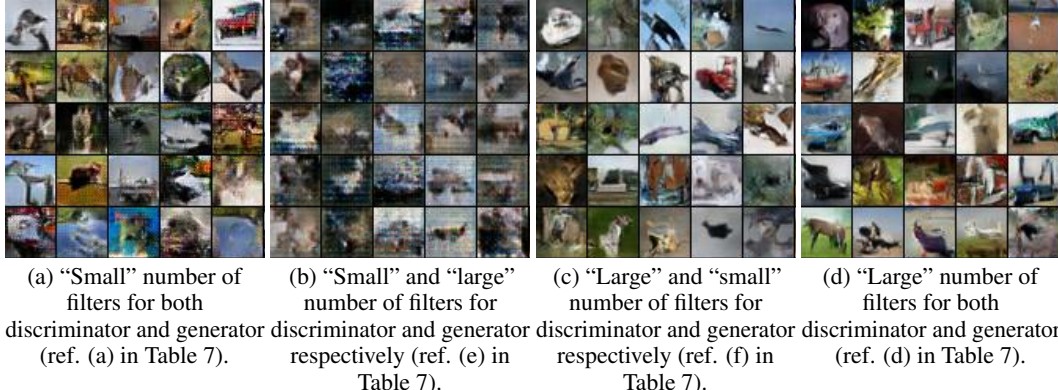

(a) "Small" number of filters for both discriminator and generator (ref. (a) in Table 7).

(b) "Small" and "large" number of filters for discriminator and generator respectively (ref. (e) in Table 7).

(c) "Large" and "small" number of filters for discriminator and generator respectively (ref. (f) in Table 7).

(d) "Large" number of filters for both discriminator and generator (ref. (d) in Table 7).

Figure 18: Samples from different architectures of LS-DCGAN

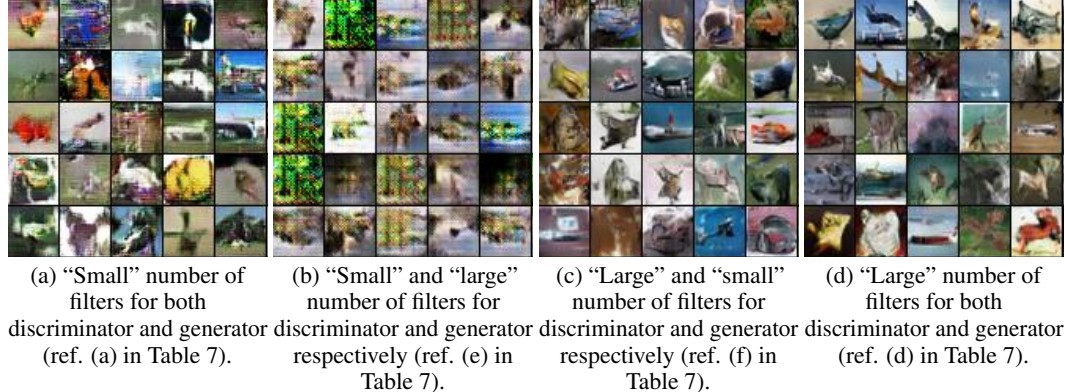

(a) "Small" number of filters for both discriminator and generator (ref. (a) in Table 7).

(b) "Small" and "large" number of filters for discriminator and generator respectively (ref. (e) in Table 7).

(c) "Large" and "small" number of filters for discriminator and generator respectively (ref. (f) in Table 7).

(d) "Large" number of filters for both discriminator and generator (ref. (d) in Table 7).

Figure 19: Samples from different architectures of W-DCGAN.

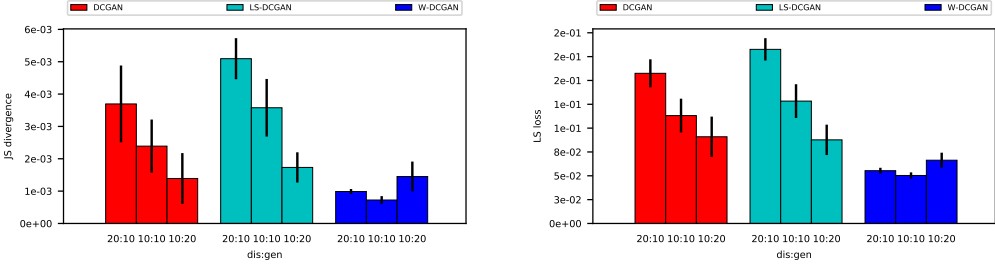

Figure 20: The performance of GANs trained with different numbers of feature maps.

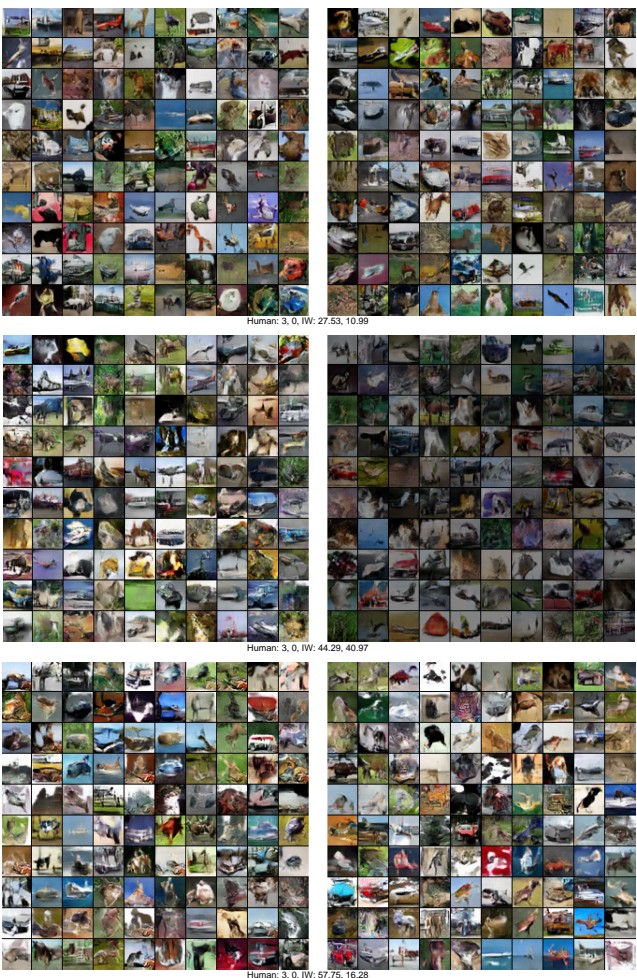

Figure 21: All pairs of generated image sets for which human perception and IW disagree, as in Figure 3.

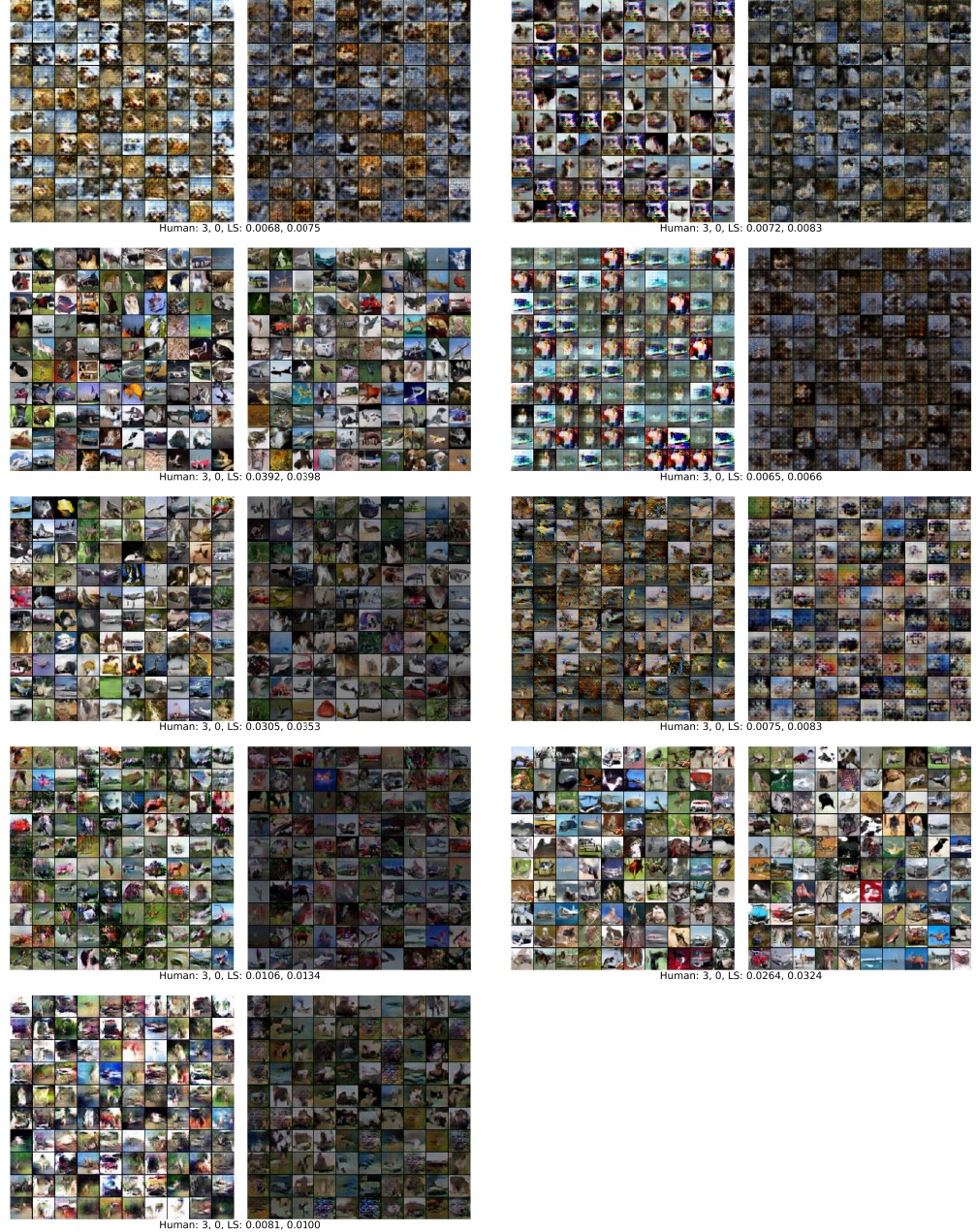

Figure 22: All pairs of generated image sets for which human perception and LS disagree, as in Figure 3.

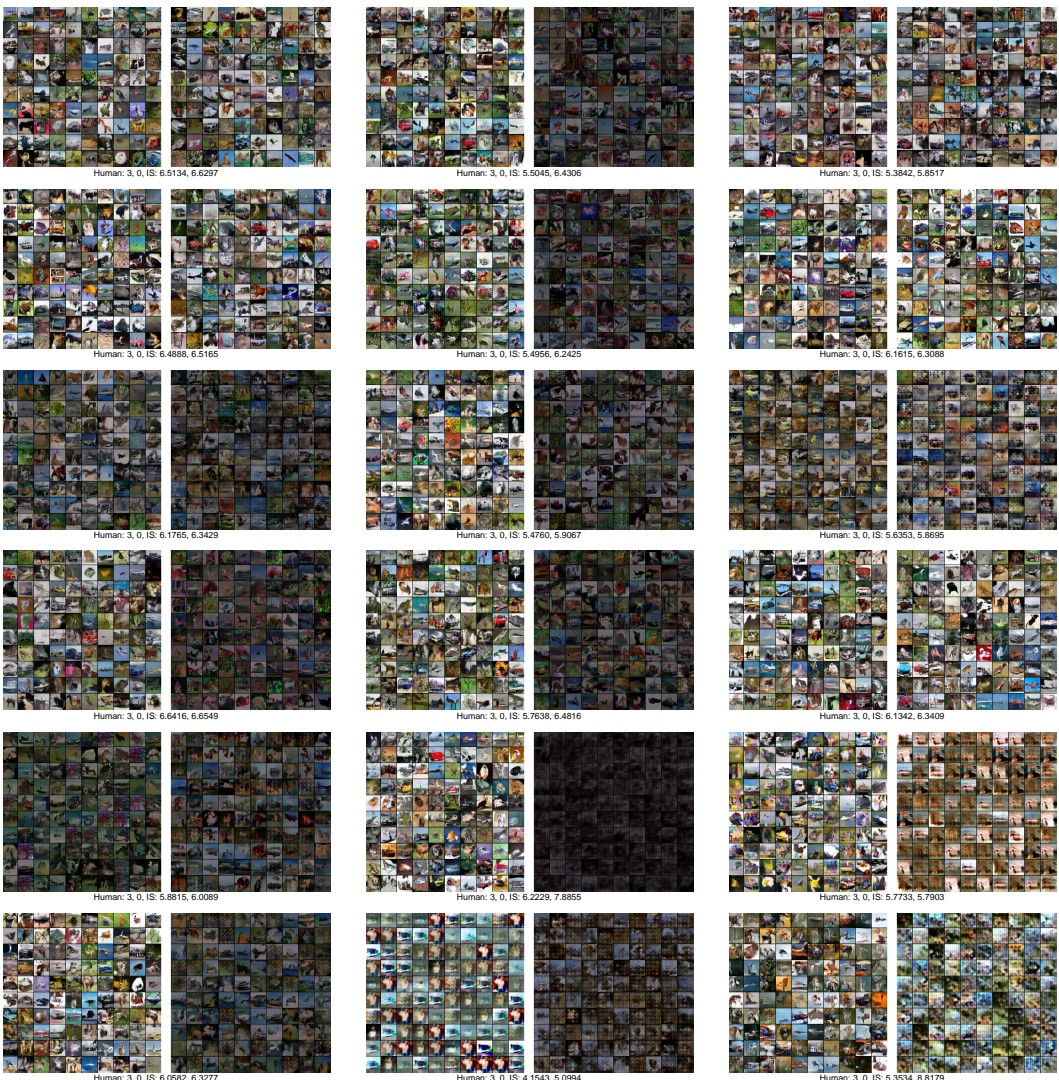

Figure 23: All pairs of generated image sets for which human perception and IS disagree, as in Figure 3.

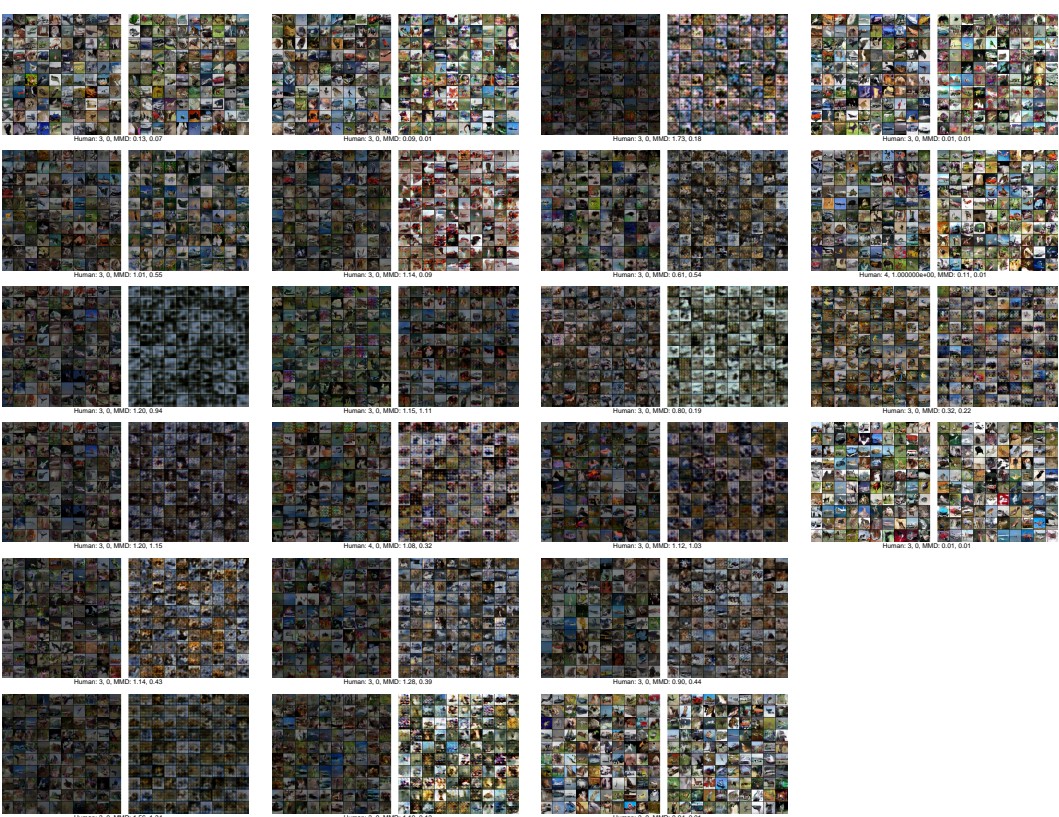

Figure 24: All pairs of generated image sets for which human perception and MMD disagree, as in Figure 3.

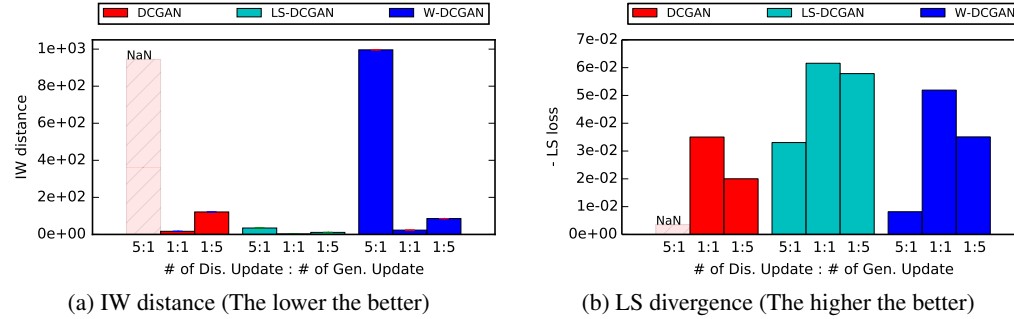

(a) IW distance (The lower the better)    (b) LS divergence (The higher the better)

Figure 25: Performance of DCGAN, W-DCGAN, and LS-DCGAN trained with varying numbers of discriminator and generator updates. These models were trained on CIFAR10 dataset and evaluated with IW and LS metrics.

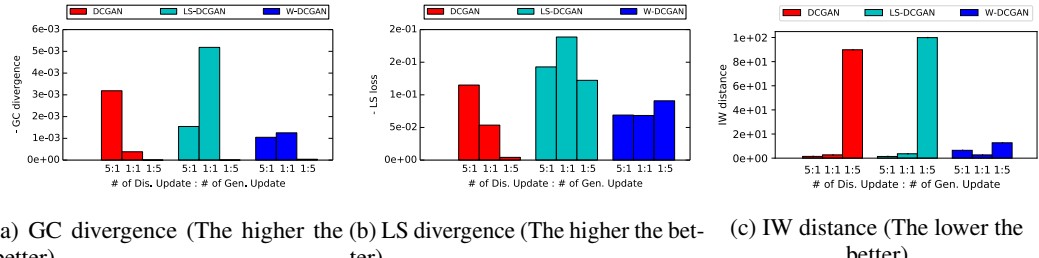

(a) GC divergence (The higher the better)    (b) LS divergence (The higher the better)    (c) IW distance (The lower the better)

Figure 26: Performance of DCGAN, W-DCGAN, and LS-DCGAN trained with varying numbers of discriminator and generator updates. These models were trained on the MNIST dataset and evaluated with GC, LS, and IW metrics.

| Ratio | DCGAN Samples |
|---|---|
| 1:1 | |
| 1:5 | |

| Ratio | W-DCGAN Samples |
|---|---|
| 5:1 | |
| 1:1 | |
| 1:5 | |

| Ratio | LS-DCGAN Samples |
|---|---|
| 5:1 | |
| 1:1 | |
| 1:5 | |

Figure 27: Samples at varying update ratios.

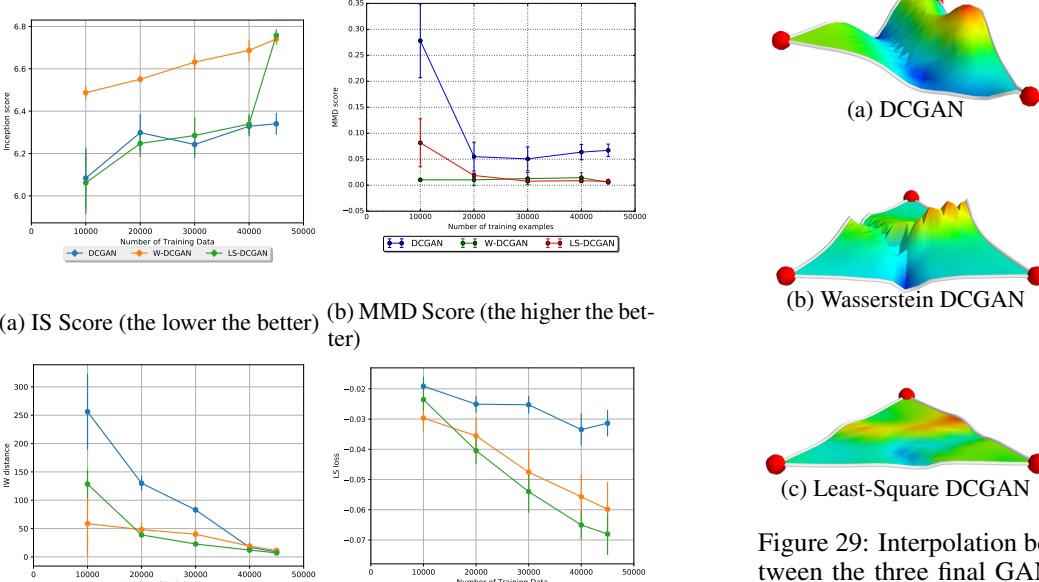

(a) IS Score (the lower the better)

(b) MMD Score (the higher the better)

(c) IW Score (the lower the better)

(d) LS Score (the higher the better)

Figure 28: Performance of W-DCGAN & LS-DCGAN with respect to number of data points.

(a) DCGAN

(b) Wasserstein DCGAN

(c) Least-Square DCGAN

Figure 29: Interpolation between the three final GAN parameters trained using different random seeds on CIFAR10. Loss surface values are amplified by 10 times in order to illustrate the separation of the terrains. Local zig-zag patterns are minor artifacts from rendering.

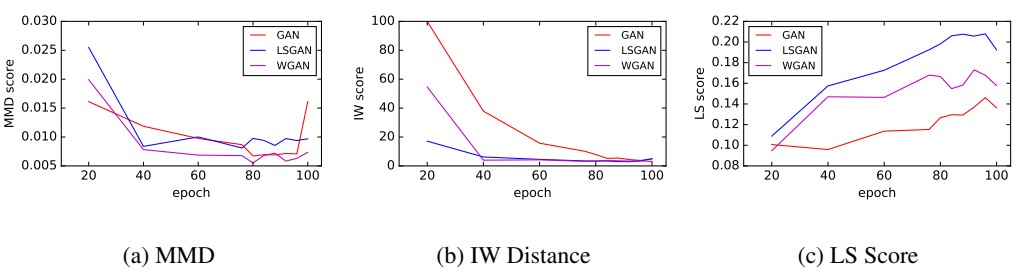

(a) MMD

(b) IW Distance

(c) LS Score

Figure 30: Scores from training GANs on LSUN Bedroom dataset.

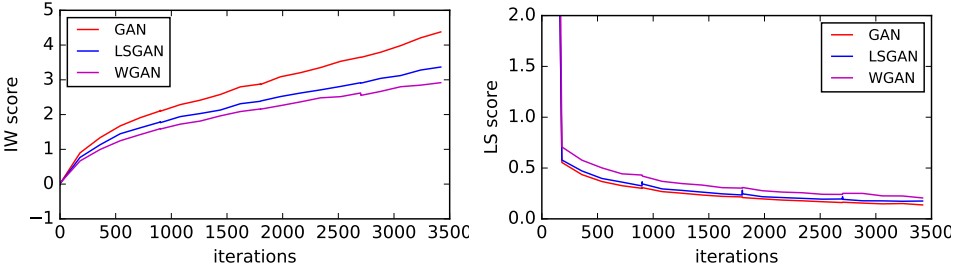

(a) The critic training curve for IW distance.

(b) The critic training curve for LS score.

Figure 31: The training curve of critics to show that the training curve converges. IW distance curves in (a) increase because we used linear output unit for the critic network (by design choice). This can be simply bounded by adding a sigmoid at the output of the critic network.

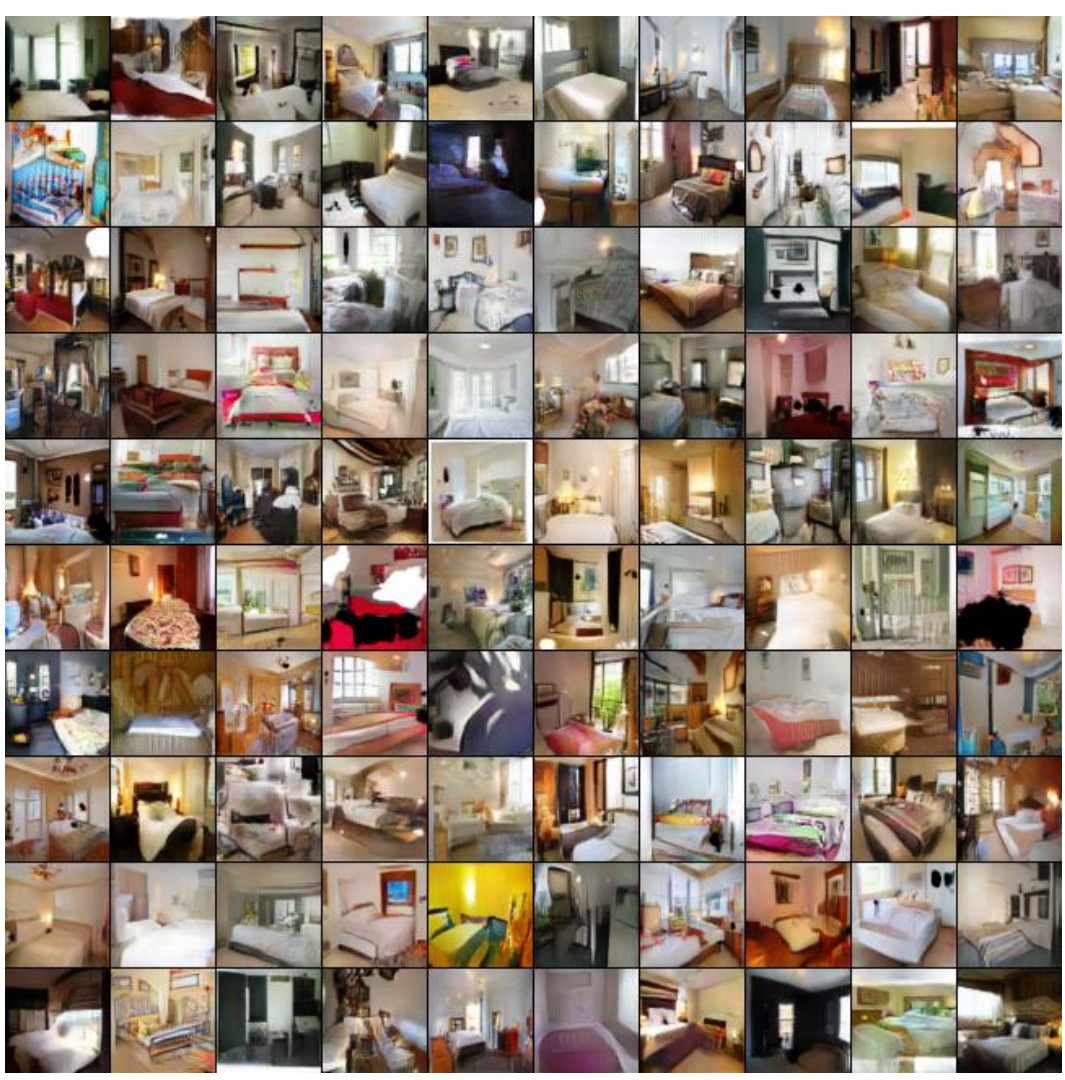

(a) GAN samples of LSUN Bedroom dataset.

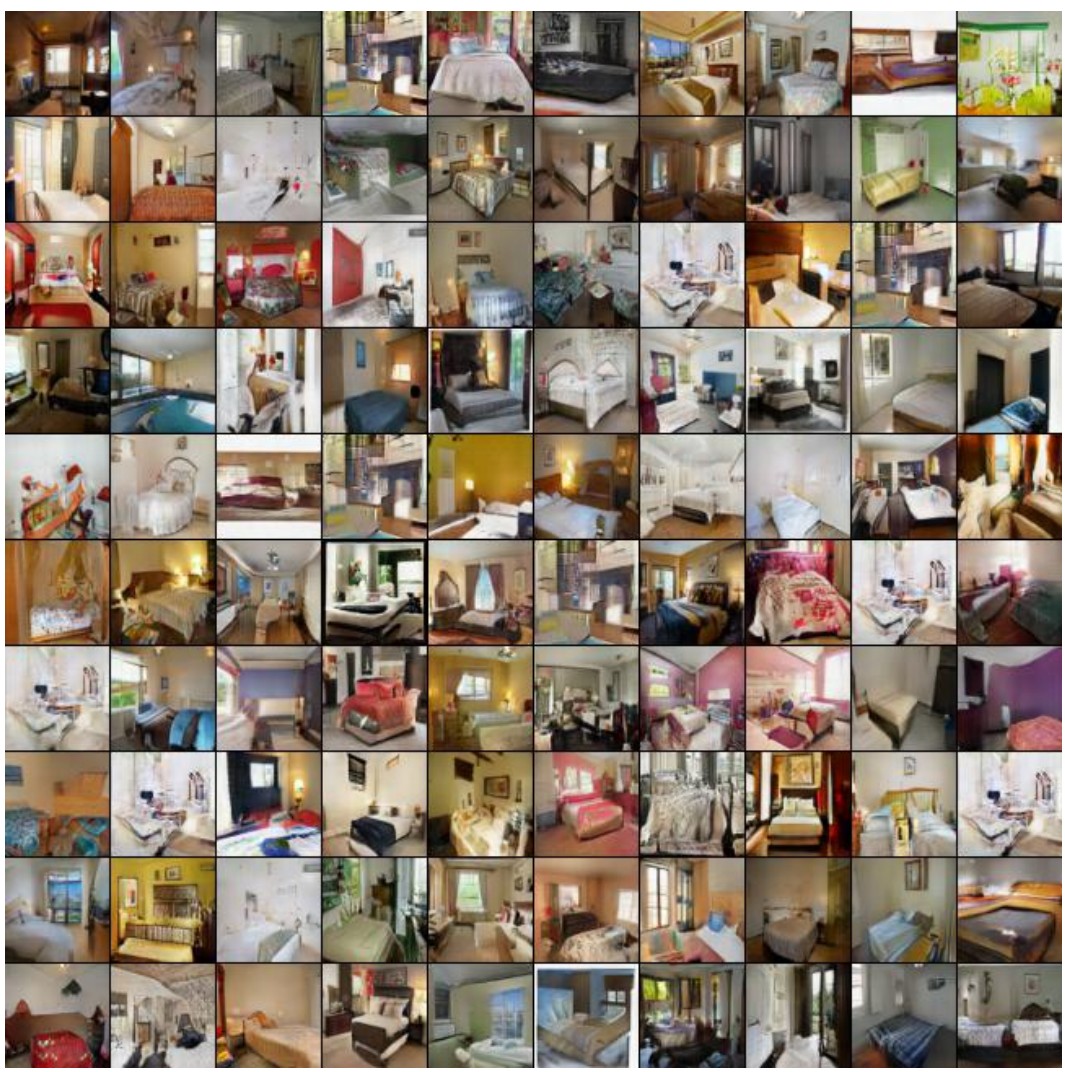

(a) LS-DCGAN samples of LSUN Bedroom dataset.

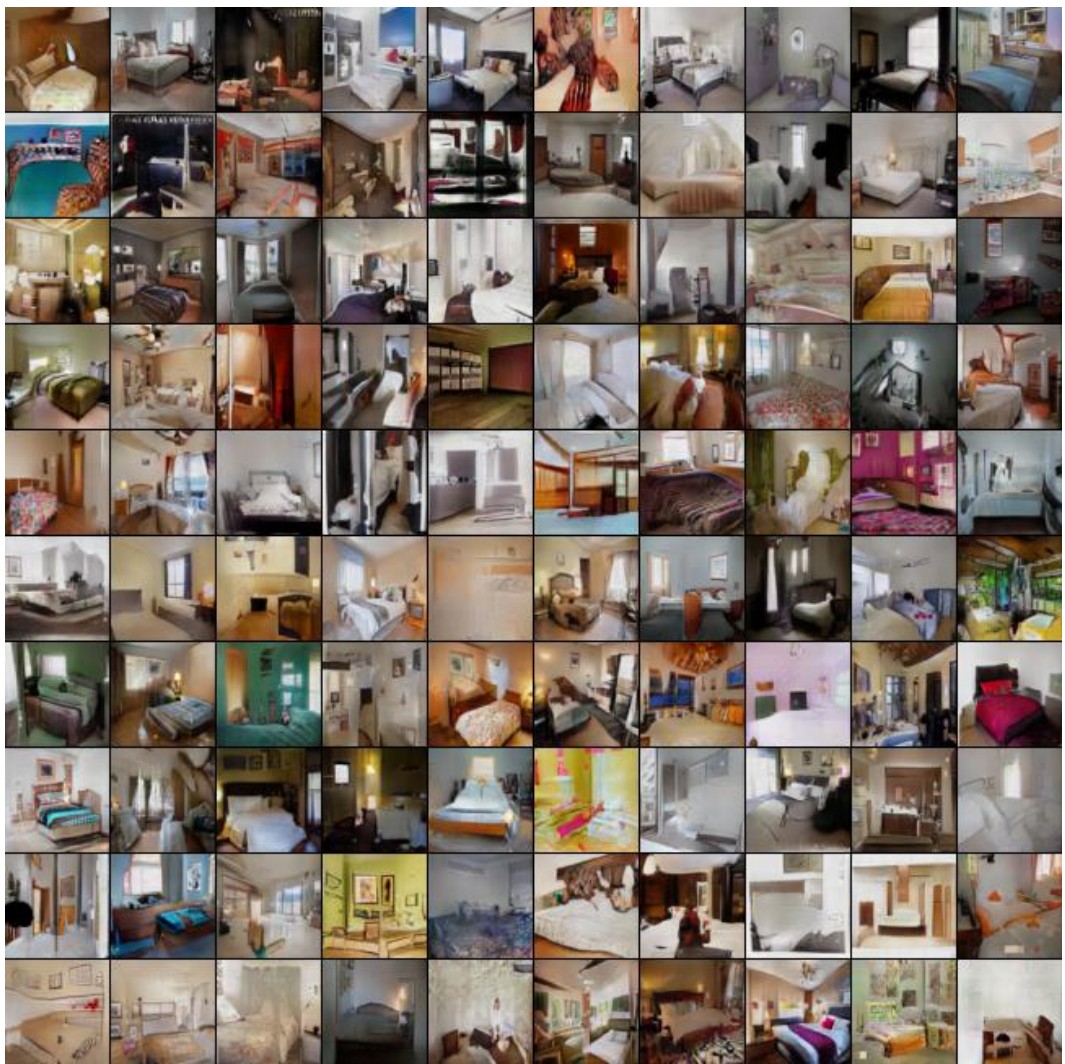

(a) W-DCGAN samples of LSUN Bedroom dataset.

