# OpenReview forum: "Quantitatively Evaluating GANs With Divergences Proposed for Training"
_ICLR.cc/2018/Conference — Accept (Poster)_

### Official Review · AnonReviewer2 · 2017-11-20
**Through evaluation of current popular GAN variants.**

**Rating:** 7
**Confidence:** 5

**Review:**

Through evaluation of current popular GAN variants.
  * useful AIS figure
  * useful example of failure mode of inception scores
   * interesting to see that using a metric based on a model’s distance does not make the model better at that distance
the main criticism that can be given to the paper is that the proposed metrics are based on trained models which do not have an independent clear evaluation metrics (as classifiers do for inception scores). However, the authors do show that the results are consistent when changing the critic architecture. Would be nice to see if this also holds for changes in learning rates.
 * nice to see an evaluation on how models scale with the increase in training data.

Using an Independent critic for evaluation has been proposed and used in practice before, see “Comparison of Maximum Likelihood and GAN-based training of Real NVPs”, Danihelka et all, as well as Variational Approaches for Auto-Encoding Generative Adversarial Networks, Rosca at all.

Improvements to be added to the paper:
   * How about overfitting? Would be nice to mention whether the proposed metrics are useful at detecting overfitting. From algorithm 1 one can see that the critic is trained on training data, but at evaluation time test data is used. However, if the generator completely memorizes the training set, the critic will not be able to learn anything useful. In that case, the test measure will not provide any information either. A way to go around this is to use validation data to train the critic, not training data. In that case, the critic can learn the difference between training and validation data and at test time the test set can be used.
  * Using the WGAN with weight clipping is not a good baseline. The improved WGAN method is more robust to hyper parameters and is the one currently  used by the community. The WGAN with weight clipping is quite sensitive to the clipping hyperparameter, but the authors do not report having changed it from the original paper, both for the critic or for the discriminator used during training.
  *  Is there a guidance for  which metric should be used?

Figure 3 needs to be made a bit larger, it is quite hard to read in the current set up.

---

> ### Author Response · Authors · 2018-01-05
> **Thank you for your review**
>
> Thank you for your review and thank you for directing us to “Comparison of Maximum Likelihood and GAN-based training of Real NVPs” by Danihelka et al. We have included references to this work in the revised paper.
>
> Regarding the comment on WGAN with weight clipping :
> We agree with the reviewer. We will include the clipping hyperparameter, which was 0.1. Additionally, we added experiments with the (improved) WGAN with gradient penalty as shown above. The results can be also found in the updated version of the paper.
>
>
>
> Regarding the comment on overfitting :
> We agree with the reviewer’s comments. As such, we experimented with the scenario proposed by the reviewer. We trained two critics on training data and validation data respectively and evaluated on test data from both critics.
>
> We trained six GANs (GAN, LSGAN, WGAN_GP, DRAGAN, BEGAN, EBGAN) on MNIST and Fashion MNIST.  We trained these GANs with 50K training examples. At test time, we used 10k training and 10k validation examples for training the critics, and evaluated on 10k test examples. Here, we present the test scores from the critics trained on training and validation data :
>
> Fashion-MNIST
>              LS score (trained on training data)  LS score (trained on validation data)
> LSGAN          0.135 +- 0.0046                           0.136 +- 0.0074
> GAN 	      0.1638 +- 0.010                           0.1635 +- 0.0006
> DRAGAN      0.1638 +- 0.015                           0.1645 +- 0.0151
> BEGAN         0.1133 +- 0.042                           0.0893 +- 0.0095
> EBGAN         0.0037 +- 0.0009                         0.0048 +- 0.0023
> WGAN_GP   0.000175 +- 0.0000876               0.000448 +- 0.0000862
>
> MNIST
>              LS score (trained on training data)  LS score (trained on validation data)
> LSGAN          0.323 +- 0.0104                          0.352 +- 0.0143
> GAN 	      0.312 +- 0.010                            0.4408 +- 0.0201
> DRAGAN      0.318 +- 0.012                             0.384 +- 0.0139
> BEGAN         0.081 +- 0.016                             0.140 +- 0.0329
> EBGAN         3.38e-6 +- 0.1.86e-7                   3.82e-6 +- 2.82e-7
> WGAN_GP   0.196 +- 0.006                             0.307 +- 0.0381
>
> Note that we also have the IW and FID evaluation on these models in the paper. For Fashion-MNIST, we find that test scores with critic trained on training and validation data are very close. Hence, we don’t see any indication of overfitting. On the other hand, there are gaps between the scores for the MNIST dataset and the test scores from critics trained on validation set gives better performance than the ones that are trained on the training set.
>
>
>
> Regarding the comment on guidance towards a metric :
>
> In terms of guidance on which metric to use, we recommend using the metric that is the closest to the human perceptual score! Thus, we added a comparison between the evaluation metrics to human perceptual scores. Please see the response (***) to Reviewer 1.
>
>
> Thank you!!

---

### Official Review · AnonReviewer1 · 2017-11-28

**Rating:** 4
**Confidence:** 3

**Review:**

This paper proposes using divergence and distance functions typically used for generative model training to evaluate the performance of various types of GANs. Through numerical evaluation, the authors observed that the behavior is consistent across various proposed metrics and the test-time metrics do not favor networks that use the same training-time criterion.

More specifically, the evaluation metric used in the paper are: 1) Jensen-Shannon divergence, 2) Constrained Pearson chi-squared, 3) Maximum Mean Discrepancy, 4) Wasserstein Distance, and 5) Inception Score. They applied those metrics to compare three different GANs: the standard DCGAN, Wasserstein DCGAN, and LS-DCGAN on MNIST and CIFAR-10 datasets.

Summary:
——
In summary, it is an interesting topic, but I think that the paper does not have sufficient novelty. Some empirical results are still preliminary. It is hard to judge the effectiveness of the proposed metrics for model selection and is not clear that those metrics are better qualitative descriptors to replace visual assessment. In addition, the writing should be improved. See comments below for details and other points.

Comments:
——
1.	In Section 3, the evaluation metrics are existing metrics and some of them have already been used in comparing GAN models.  Maximum mean discrepancy has been used before in work by Yujia Li et al. (2016, 2017)

2.	In the experiments, the proposed metrics were only tested on small scale datasets; the authors should evaluate on larger datasets such as CIFAR-100, Toronto Faces, LSUN bedrooms or CelebA.

3.	In the experiments, the authors noted that “Gaussian observable model might not be the ideal assumption for GANs. Moreover, we observe a high log-likelihood at the beginning of training, followed by a drop in likelihood, which then returns to the high value, and we are unable to explain why this happens.” Could the authors give explanation to this phenomenon? The authors should look into this more carefully.

4.	In algorithm 1, it seems that the distance is computed via gradient decent. Is it possible to show that the optimization always converges? Is it meaningful to compare the metrics if some of them cannot be properly computed?

5.     With many different metrics for assessing GANs, how should people choose? How do we trust the scores? Recently, Fréchet Inception Distance (FID) was proposed to evaluate the samples generated from GANs (Heusel et al. 2017), how are the above scores compared with FID?

Minor Comments:
——
1.	Writing should be fixed: “It seems that the common failure case of MMD is when the mean pixel intensities are a better match than texture matches (see Figure 5), and the common failure cases of IS happens to be when the samples are recognizable textures, but the intensity of the samples are either brighter or darker (see Figure 2).”

---

> ### Author Response · Authors · 2018-01-05
> **Thank you for your review!**
>
> Thank you for your review!
>
> Regarding comment 1 :
> The reviewer noted that MMD has been used in previous works by Yujia Li et al. [1,2]. Indeed, MMD has been proposed as a training objective in many previous works. Nevertheless, the goal of this paper was to consider different evaluation metrics for scoring GANs and test whether one type of GANs is statistically better than the other one under different metrics. Our claim of novelty is not in proposing a new metric but in evaluating GANs under many different metrics. We made an observation that many metrics have been used to train GANs but surprisingly have not been used to evaluate GANs at test time. Hence, we used those metrics with a critic network to evaluate GANs. Li et al. [1,2] have not employed MMD as a GAN evaluation metric at test time. In the paper, we systematically compared and ranked GANs under different metrics.
>
>
> Regarding comment 2 : Please see our response to Reviewer 3.
>
>
> Regarding comment 3 : The mentioned phenomenon for log-likelihood using AIS by [3] is interesting. However, we do not know why it has that behaviour and we believe that this is not within the scope of our paper to find out. It would be best to directly ask the authors of [3].
>
> Regarding comment 4: The reviewer asked about convergence guarantees using gradient descent. Note that gradient descent is employed widely in deep learning and optimizing the critic’s objective (the distance) is exactly same as training a deep feedforward network with gradient descent. The scope of this comment falls under all deep learning methods and it is not specific to our paper.
>
> Note : We include the training curve of critics to show that at least the training curve converges (see Figure 26).
>
> Regarding comment 5: Please see our response to the review (Fréchet Inception Distance for evaluating GANs) from Nov. 20th 2017. We have included substantial experimental results in the updated version of the paper.
>
> (***)
> Moreover, we added a comparison between the evaluation metrics and human perceptual scores. We showed which metrics are more statistically correlated with human perceptual scores.  This was done based on the Wilcoxon rank sum test & two-sided Fisher’s test. The fraction of pairs on which each metric agrees with humans (higher the better):
>
>       Metric                      Fraction Agreed                  agreed # samples / # total samples
> Inception score              0.862595                                            ( 113 / 131)
> IW Distance                    0.977099                                            (128 / 131)
> LS score                           0.931298                                            (122 / 131)
> MMD                                0.83261                                              (109 / 131)
>
>
> It shows that IW distance agreed the most with human perceptual scores, and then followed by LS, Inception score, and MMD.
>
> Also, here are the results of a two-sided Fisher’s test (no multiple comparison correlation) that these fractions are the same or not:
>
> IS equals IW      : p-value = 0.000956
> IS equals LS      : p-value = 0.102621
> IS equals MMD  : p-value = 0.606762
> IW equals LS     : p-value = 0.136684
> IW equals MMD : p-value = 0.000075
> LS equals MMD : p-value = 0.020512
>
> Over all, it demonstrates that IW, LS are significantly more aligned with the perception scores than the Inception Score and MMD, p < 0.05. (See the Section 4.2.1 for details).
>
> We have improved the quality of the writing in the revised paper.
> Thank you!!
>
>
> [1] Yujia Li, Kevin Swersky and Richard Zemel.  Generative Moment Matching Networks. International Conference on Machine Learning (ICML), 2015
> [2] Yujia Li, Alexander Schwing, Kuan-Chieh Wang, Richard Zemel. Dualing GANs. https://arxiv.org/abs/1706.06216
> [3] Yuhuai Wu, Yuri Burda, Ruslan Salakhutdinov and Roger Grosse. On the Quantitative Analysis of Decoder-Based Generative Models.   ICLR, 2017.

---

### Official Review · AnonReviewer3 · 2017-11-30
**new evaluation metrics for GANs**

**Rating:** 7
**Confidence:** 4

**Review:**

the paper proposes an evaluation method for training GANs using four standard distribution distances in literature namely:
- JSD
- Pearson-chi-square
- MMD
- Wasserstein-1

For each distance, a critic is initialized with parameters p. The critic is a neural network with the same architecture as the discriminator.
The critic then takes samples from the trained generator model, and samples from the groundtruth dataset. It trains itself to maximize the distance measure between these two distributions (trained via gradient descent).

These critics after convergence will then give a measure of the quality of the generator (lower the better).

The paper is easy to read, experiments are well thought out.
Figure 3 is missing (e) and (f) sub-figures.

When proposing a distance measure for GANs (which is a high standard, because everyone is looking forward to a robust measure), one has to have enough convincing to do. The paper only does experiments on two small datasets MNIST and CIFAR. If the paper has to convince me that this metric is good and should be used, I need to see experiments on one large-scale datset, such as Imagenet or LSUN. If one can clearly identify the good generators from bad generators using a weighted-sum of these 4 distances on Imagenet or LSUN, this is a metric that is going to stand well.
If such experiments are constructed in the rebuttal period, I shall raise my rating.

---

> ### Author Response · Authors · 2018-01-05
> **Thank you for your review.**
>
> Thank you for your review.
>
> At the reviewer’s suggestion, we conducted the same experiments on the LSUN bedroom dataset. We used 64 x 64 of 90,000 images to train GAN, LSGAN, WGAN and tested on another 90,000 unseen images. We evaluated using the LS score, IW distance, and MMD. We omitted the Inception score, because LSUN bedroom dataset contains just a single class and there is no pre-trained convolutional network available (inception score needs a pre-trained convolutional network). Samples from each model are also added in the appendix of the paper. Here is a summary of the results:
>
>                 LS (higher the better)     IW (lower the better)    MMD (higher the better)
> GAN :     0.14614                             3.79097                         0.00708
> LSGAN:  0.173077                           3.36779                         0.00973
> WGAN :  0.205725                           2.91787                         0.00584
>
> All three metrics agrees that WGAN has the best score. LSGAN is ranked second according to the LS score and IW distance, in contrast, MMD puts GAN in second place and LSGAN on third place. Nevertheless, in our more recent experiments added to the revised version of the paper, we showed that MMD score often disagrees with human perceptual score.
>
> In summary, we applied our evaluation methods to larger images and the performance of IW and LS are consistent with what we observed on MNIST and CIFAR10.
>
> We added these results to the paper.
>
>
> Additionally, we would like to note that we added a comparison between the evaluation metrics to human perceptual scores. Please see the response (***) to Reviewer 1.
>
> Thank you!!

---

### Public Comment · (anonymous) · 2017-10-30
**C2ST**

What is the relationship between the methods proposed here and classifier two-sample tests?

https://arxiv.org/abs/1610.06545

---

> ### Author Response · Authors · 2017-11-11
> **Thank you for directing us to C2ST**
>
> Thank you for directing us to C2ST.
>
> There is a relationship between the methods proposed and classifier two-sample tests (C2ST). C2ST proposes to train a classifier that can distinguish samples drawn from two distribution P and Q and accept/reject the null hypothesis.
>
> One of the commonalities that shared between our proposed test methods and C2ST is that both require optimizing a function (training a neural network) at test time. C2ST trains neural network to maximize the classification between data and samples, whereas our method is training neural network to maximize the distance between the data and samples.
>
> In our paper, we considered four distance metrics that belong to two class of metrics, $\phi$-divergence and IPMs. Sriperumbudur et al. have shown that the optimal risk function is associated with a binary classifier with P and Q distribution conditioned on class when the discriminant function is restricted to certain F (Theorem 17 from Sriperumbudur et al).
>
> Let the optimal risk function be
>
>    R(L,F) = inf_{f \in F} \int L(y, f(x)) dp(x,y)
>
> where F is the set of discriminant functions (classifier), y \in {-1,1}, and L is the loss function.
>
> By following derivation, we can see that the optimal risk function becomes IPM:
>
> R(L,F) = inf_{f \in F} \int L(y, f(x)) du(x,y)
>            = inf_{f \in F} [  eps \int L(1, f(x)) dp(x) + (1 - eps)  \int L(0, f(x)) dq(x)
>            = inf_{f \in F} f dp(x) + \inf_{f \in F} f dq(x)
>            = - IPM
>
> where L(1, f(x)) = 1/eps and L(0,f(x)) = -1 / (1-eps).
>
> The second equality is derived by separating the loss for class 1 and class 0.
> Third equality is from the way how we chose L(1,f(x)) and L(0,f(x)).
> The last equality follow by the fact that F is symmetric around zero (f \in F => -f \in F). Hence, this shows that with appropriately choosing L, MMD, Wasserstein distance can be understood as the optimal L-risk associated with binary classifier with specific set of F functions. For example, Wasserstein distance and MMD distances are equivalent to optimal risk function with 1-Lipschitz classifiers and RKHS classifier with an unit length.
>
> Similarly, since every binary classifier has a corresponding distance metric from IPM, C2ST binary classifier must have a distance function associated with it with specific set of F. For example, if the binary classifier is KNN, then we are considering IPM metric with the topology induced by KNN, as well if the classifier is neural networks then we are considering IPM metric with the topology induced by the neural network.
>
> Thank you for asking relationship between our proposed testing methods and C2ST.
> We will make sure to add these descriptions in the paper for the next revision cycle!!!
>
>
> [1] Sriperumbudur et al.  On Integral Probability Metrics, $\phi$-divergence and binary classification.

---

### Public Comment · (anonymous) · 2017-11-14
**Fréchet Inception Distance (FID) for evaluating GANs**

[1] proposed the Fréchet Inception Distance (FID) to evaluate GANs which is the Fréchet distance aka Wasserstein-2 distance between the real world and generated samples statistics.  [1] showed in their experiments clearly a much more  consistent behaviour of the FID compared to the Inception Score. It is now unclear if the analysis in this paper could be improved by using the FID for GAN evaluations.

[1] https://arxiv.org/abs/1706.08500

---

> ### Author Response · Authors · 2017-11-21
> **Thank you for directing us to FID paper.**
>
> Thank you for directing us to FID paper.
>
> As the reviewer stated, FID computes the distance between the Gaussian in Wasserstein-2 distance. The sufficient statistics of Gaussian comes from the first and second moment of the neural network features (convolutional neural network's feature maps).
>
> Compare to our proposed evaluation methods, FID has an advantage and disadvantage. The main advantage is the speed. The calculating of FID distance is much faster than our evaluation methods. The disadvantage of FID is that it only considers difference in first two moments of the samples, which can be insufficient unless the feature maps are Gaussian distributed. On the other hand, the four metrics that we consider does not make any assumptions about the distribution of the samples.
>
> We agree with the reviewer on analyzing the experiments with FID! So, we have run experiments that includes FID:  We included the FID scores to Table 3, which shows the overall performance of DCGAN, LSGAN, WGAN on CIFAR10. We also evaluated on more recently proposed models, such as DRAGAN, BEGAN, EBGAN, WGAN_GP, based on the off-the-shelf package on MNIST and Fashion-MNIST.  The evaluation metric includes, LS, IW and FID.
>
>
> For CIFAR10, the FID results agree with LS. The FID results are following:
>          CIFAR10    FID
>           DCGAN : 0.112 +- 0.010;
>           W-DCGAN : 0.095 +- 0.003;
>           LS-DCGAN : 0.088 +- 0.008
> (the smaller FID the better; see the Table 3 for other metric scores). According to FID, LS-DCGAN is the best among three models.
>
> For MNIST, the three metrics, LS, IW, and FID, agree with each other on the rank as well. They all find that samples from DRAGAN is the best, then LSGAN, and so on.
>          MNIST       Metric
>         DCGAN       IW: 0.111 +- 0.0074 LS: 0.4814 +- 0.0083  FID: 1.84 +- 0.15
>         EBGAN       IW: 0.029 +- 0.0026 LS: 0.7277 +- 0.0159  FID: 5.36 +- 0.32
>         WGAN GP  IW: 0.035 +- 0.0059 LS: 0.7314 +- 0.0194  FID: 2.67 +- 0.15
>         LSGAN        IW: 0.115 +- 0.0070 LS: 0.5058 +- 0.0117  FID: 2.20 +- 0.27
>         BEGAN       IW: 0.009 +- 0.0063 LS: -	   		            FID: 15.9 +- 0.48
>         DRAGAN    IW: 0.116 +- 0.0116 LS: 0.4632 +- 0.0247  FID: 1.09 +- 0.13
>
> For Fashion-MNIST, LS, IW and FID agree on the rank of worst ones, but there are some subtle difference between LS and IW versus FID.   According to LS and IW, DRAGAN samples are ranked first and LSGAN samples are ranked second, and visa versa for FID.
>         Fashion-MNIST       Metric
>         DCGAN       IW: 0.69 +- 0.0057  LS: 0.0202 +- 0.00242 FID:   3.23 +- 0.34
>         EBGAN       IW: 0.99 +- 0.0001  LS: 2.2e-5 +- 5.3e-5  FID: 104.08 +- 0.56
>         WGAN GP     IW: 0.89 +- 0.0086  LS: 0.0005 +- 0.00037 FID:   2.56 +- 0.25
>         LSGAN       IW: 0.68 +- 0.0086  LS: 0.0208 +- 0.00290 FID:   0.62 +- 0.13
>         BEGAN       IW: 0.90 +- 0.0159  LS: 0.0016 	  0.00047 FID:   1.51 +- 0.16
>         DRAGAN      IW: 0.66 +- 0.0108  LS: 0.0219 +- 0.00232 FID:   0.97 +- 0.14
>
>
> We will make sure to add these descriptions and experiment results in the paper for the next revision cycle.
>
> Thank you!!

---

### Public Comment · (anonymous) · 2017-12-15
**Reference suggestion: variants of C2ST to evaluate GANs**

Hi all,
I have a reference recommendation: prior work using variants of C2ST for evaluating GANs.
GANs for Biological Image Synthesis
Anton Osokin, Anatole Chessel, Rafael E. Carazo Salas, Federico Vaggi
ICCV 2017
https://arxiv.org/abs/1708.04692

This paper used Metric 1 and Metric 4 (and the approximation of Wasserstein derived from WGAN-GP) to evaluate performance of GAN, WGAN, WGAN-GP for a specific application. They did a detailed study of how these metrics were correlated with the visual quality and several sanity checks of these metrics (see Section 5.1 and appendix A). I think it would be appropriate to cite this work.

---

> ### Author Response · Authors · 2018-01-05
> **Thank you for directing us “GANs for Biological Image Synthesis”**
>
> Thank you for directing us to “GANs for Biological Image Synthesis” by Osokin et al. We have included references to this work in the revised paper.
>
> Thank you!!

---

### Author Response · Authors · 2018-01-05
**Summary to the paper update.**

The revised version of the paper contains several additional experimental contributions motivated by the reviews we received. Though we have details of each of these in our responses to the reviews, we wanted to provide a short summary as follows:

- Comparison with human perceptual scores (Correlation test between different metrics to human perceptual scores)
- Evaluation on larger image data, LSUN bedrooms
- Use of the (improved) WGAN with gradient penalty
- Comparison to Fréchet Inception Distance
- Investigating the metrics as a means of detecting overfitting

Thank you for your interest!

---

### Decision · Program_Chairs · 2018-01-29
**ICLR 2018 Conference Acceptance Decision**

**Decision:**

Accept (Poster)

**Comment:**

+ clearly written and thorough empirical comparison of several metrics/divergences for evaluating GANs, prominently parametric-critic based divergences.
 - little technical novelty with respect to prior work. As noted by reviewers and an anonymous commentator:  using an Independent critic for evaluation has been proposed and used in practice before.
 +  the contribution of the work thus lies primarily in its well-done and extensive empirical comparisons of multiple metrics and models